# Towards Robust Multimodal Sentiment Analysis with Incomplete Data

**Haoyu Zhang**[1,2]**, Wenbin Wang**[3]**, Tianshu Yu**[1,*]
[1]School of Data Science, The Chinese University of Hong Kong, Shenzhen
[2]Department of Computer Science, University College London
[3]School of Computer Science, Wuhan University
{zhanghaoyu, yutianshu}@cuhk.edu.cn
haoyu.zhang.23@ucl.ac.uk
wangwenbin97@whu.edu.cn

## Abstract

The field of Multimodal Sentiment Analysis (MSA) has recently witnessed an emerging direction seeking to tackle the issue of data incompleteness. Recognizing that the language modality typically contains dense sentiment information, we consider it as the dominant modality and present an innovative Language-dominated Noise-resistant Learning Network (LNLN) to achieve robust MSA. The proposed LNLN features a dominant modality correction (DMC) module and dominant modality based multimodal learning (DMML) module, which enhances the model's robustness across various noise scenarios by ensuring the quality of dominant modality representations. Aside from the methodical design, we perform comprehensive experiments under random data missing scenarios, utilizing diverse and meaningful settings on several popular datasets (*e.g.,* MOSI, MOSEI, and SIMS), providing additional uniformity, transparency, and fairness compared to existing evaluations in the literature. Empirically, LNLN consistently outperforms existing baselines, demonstrating superior performance across these challenging and extensive evaluation metrics.[1]

## 1 Introduction

The field of Multimodal Sentiment Analysis (MSA) is at the vanguard of human sentiment identification by assimilating heterogeneous data types, such as video, audio, and language. Its applicability spans numerous fields, prominent among healthcare and human-computer interaction (Jiang et al., 2020). MSA has become essential in enhancing both the precision and the robustness of sentiment analysis by drawing sentiment cues from diverse perspectives.

Changing trends in recent research has taken a turn towards modeling data in natural scenarios from laboratory conditions (Tsai et al., 2019; Hazarika et al., 2020; Yu et al., 2021; Zhang et al., 2023). The shift has created a wider application space in the real-world for MSA, though concerns arise due to problems like sensor failures and problems with Automatic Speech Recognition (ASR), leading to inconsistencies such as incomplete data in real-world deployment.

Numerous impactful solutions have been proposed against this primary concern of incomplete data in multimodal sentiment analysis. For instance, Yuan et al. (2021) introduced a Transformer-based feature reconstruction mechanism, TFR-Net, aiming to strengthen the robustness of the model handling random missing in unaligned multimodal sequences via reconstructing missing data.

---

[*]the corresponding author
[1]The code is available at: `https://github.com/Haoyu-ha/LNLN`

38th Conference on Neural Information Processing Systems (NeurIPS 2024).

Furthermore, Yuan et al. (2024) proposed the Noise Intimating-based Adversarial Training (NIAT) model, which superiorly learns a unified joint representation between an original-noisy instance pair, utilizing the attention mechanism and adversarial learning. Lastly, Li et al. (2024) design a Unified Multimodal Missing Modality Self-Distillation Framework (UMDF) which leverages a single network to learn robust inherent representations from consistent multimodal data distributions. Yet, despite these developments, the evaluation metrics for these models are inconsistent, and the evaluation settings are not sufficiently comprehensive. This inconsistency limits effective comparisons and hinders the dissemination of knowledge in the field.

Addressing this gap, our paper aims to offer a comprehensive evaluation on three widely-used datasets, namely MOSI (Zadeh et al., 2016), MOSEI (Zadeh et al., 2018) and SIMS (Yu et al., 2020) datasets. We introduce random data missing instances and subsequently compare the performance of existing methods on these datasets. This endeavor seeks to provide an all-encompassing outlook for evaluating the effectiveness and robustness of various methods in the face of incomplete data, thereby sparking new insights in the field. Additionally, inspired by the previous work ALMT (Zhang et al., 2023), we hypothesize that model robustness improves when the integrity of the dominant modality is preserved despite varying noise levels. Therefore, we introduce a novel model, namely Language-dominated Noise-resistant Learning Network (LNLN), to enhance MSA's robustness over incomplete data. LNLN aims to augment the integrity of the language modality's features, regarded as the dominant modality due to its richer sentiment cues, with the support of other auxiliary modalities. The LNLN's robustness against varying levels of data incompleteness is achieved through a dominant modality correction (DMC) module for dominant modality construction, a dominant modality based multimodal learning (DMML) module for multimodal fusion and classification, and a reconstructor for reconstructing missing information to shield the dominate modality from noise interference. This approach ensures a high-quality dominant modality feature, which significantly bolsters the robustness of LNLN under diverse noise conditions. Consequently, extensive experimental results demonstrate the LNLN's superior performance across these challenging and evaluation metrics.

In summary, this paper conducts a comprehensive evaluation of existing advanced MSA methods. This analysis highlights the strengths and weaknesses of various methods when contending with incomplete data. We believe, it can improve the understanding of different MSA methods' performance under complex real-world scenarios, thereby informing technology's future trajectory. Our proposed LNLN also offers valuable insight and guidelines for thriving in this research space.

## 2 Related Work

### 2.1 Multimodal Sentiment Analysis

Multimodal Sentiment Analysis (MSA) methods can be categorized into Context-based MSA and Noise-aware MSA, depending on the modeling approach. Most of previous works (Zadeh et al., 2017; Tsai et al., 2019; Mai et al., 2020; Hazarika et al., 2020; Liang et al., 2020; Rahman et al., 2020; Yu et al., 2021; Han et al., 2021; Lv et al., 2021; Yang et al., 2022; Guo et al., 2022; Zhang et al., 2023) can be classified to Context-based MSA. This line of work primarily focuses on learning unified multimodal representations by analyzing contextual relationships within or between modalities. For example, Zadeh et al. (2017) explore computing the relationships between different modalities using the Cartesian product. Tsai et al. (2019) utilize pairs of Transformers to model long dependencies between different modalities. Yu et al. (2021) propose generating pseudo-labels for each modality to further mine the information of consistency and discrepancy between different modalities.

Despite these advances, context-based methods are usually suboptimal under varying levels of noise effects (*e.g.* random data missing). Several recent works (Mittal et al., 2020; Yuan et al., 2021, 2024; Li et al., 2024) have been proposed to tackle this issue. For example, Mittal et al. (2020) introduce a modality check step to distinguish invalid and valid modalities, achieving higher robustness. Yuan et al. (2024) propose learning a unified joint representation between constructed "original-noisy" instance pairs. Although there have been some advances in improving the model's robustness under noise scenarios, no extant method has provided a comprehensive and in-depth comparative analysis.

## 2.2 Robust Representation Learning in MSA

Context-based MSA and Noise-aware MSA differ in their approaches to robust representation learning. In Context-based MSA, robust representation learning typically relies on modeling intra- and inter-modality relationships. For instance, Hazarika et al. (2020) and Yang et al. (2022) apply feature disentanglement to each modality, modeling multimodal representations from multiple feature subspaces and perspectives. Yu et al. (2021) and Liang et al. (2020) explore self-supervised learning and semi-supervised learning to enhance multimodal representations, respectively. Tsai et al. (2019) and Rahman et al. (2020) introduce Transformer to learn the long dependencies of modalities. Zhang et al. (2023) devise a language-guided learning mechanism that uses modalities with more intensive sentiment cues to guide the learning of other modalities. In contrast, Noise-aware MSA focuses more on perceiving and eliminating the noise present in the data. For example, Mittal et al. (2020) design a modality check module based on metric learning and Canonical Correlation Analysis (CCA) to identify the modality with greater noise. Yuan et al. (2021) design a feature reconstruction network to predict the location of missing information in sequences and reconstruct it. Yuan et al. (2024) introduce adversarial learning (Goodfellow et al., 2014) to perceive and generate cleaner representations.

In this work, the LNLN belongs to the noise-aware MSA category. Inspired by Zhang et al. (2023), we explore the capability of language-guided mechanisms in resisting noise and aim to provide new perspectives for the study of MSA in noisy scenarios.

# 3 Method

## 3.1 Overview

The overall pipeline for the proposed Language-dominated Noise-resistant Learning Network (LNLN) in robust multimodal sentiment analysis is illustrated in Figure 1. As depicted, a crucial initial step involves forming a multimodal input with random data missing, which sets the stage for LNLN training. Once the input is prepared, LNLN first utilizes an embedding layer to standardize the dimension of each modality, ensuring uniformity. Recognizing that language is the dominant modality in MSA (Zhang et al., 2023), a specially designed Dominant Modality Correction (DMC) module employs adversarial learning and a dynamic weighted enhancement strategy to mitigate noise impacts. This module first enhances the quality of the dominant feature computed from the language modality and then integrates them with the auxiliary modalities (visual and audio) in the dominant modality based multimodal learning (DMML) module for effective multimodal fusion and classification. This process significantly bolsters LNLN's robustness against various noise levels. Moreover, to refine the network's capability for fine-grained sentiment analysis, a simple reconstructor is implemented to reconstruct missing data, further enhancing the system's robustness.

## 3.2 Input Construction and Multimodal Input

Given the challenges of random data missing, we have constructed data sets that simulate these conditions based on MOSI, MOSEI, and SIMS datasets.

**Random Data Missing.** Following the previous method (Yuan et al., 2021), for each modality, we randomly erased changing proportions of information (from 0% to 100%). Specifically, for visual and audio modalities, we fill the erased information with zeros. For language modality, we fill the erased information with [UNK] which indicates the unknown word in BERT (Kenton and Toutanova, 2019).

**Multimdoal Input.** For each sample in the dataset, we incorporate data from three modalities: language, audio, and visual data. Consistent with previous works (Zhang et al., 2023), each modality is processed using widely-used tools: language data is encoded using BERT (Kenton and Toutanova, 2019), audio features are extracted through Librosa (McFee et al., 2015), and visual features are obtained using OpenFace (Baltrusaitis et al., 2018). These pre-processed inputs are represented as sequences, denoted by $U_m^0 \in \mathbb{R}^{T_m \times d_m}$, where $m \in \{l, v, a\}$ represents the modality type ($l$ for language, $v$ for visual, $a$ for audio), $T_m$ indicates the sequence length and $d_m$ refers to the dimension of each modality's vector. With obtained $U_m^0$, we apply random data missing to $U_m^0$, thus forming the noise-corrupted multimodal input $U_m^1 \in \mathbb{R}^{T_m \times d_m}$.

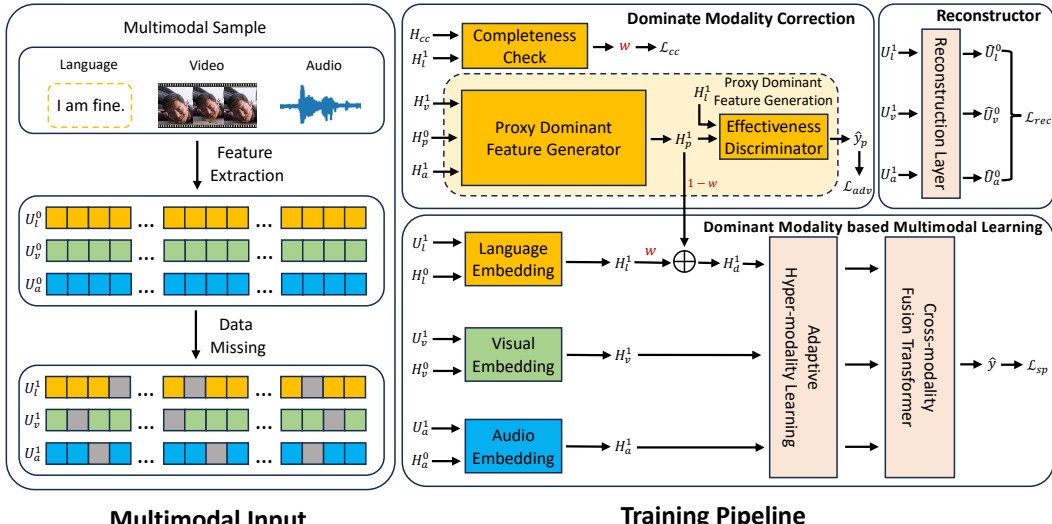

Figure 1: Overall pipeline. Note: $H_l^0$, $H_v^0$, $H_a^0$, $H_{cc}$, and $H_p^0$ are randomly initialized learnable vectors.

## 3.3 Dominant Modality based Multimodal Learning

Inspired by previous work ALMT, we hypothesize that model robustness improves when the integrity of the dominant modality is preserved despite varying noise levels. We improve ALMT based on the designed DMC module and Reconstructor, thus implementing dominant modality based DMML module for sentiment analysis under random data missing scenarios. Here, we mainly introduce the parts that differ from ALMT. Further details are available in Zhang et al. (2023).

**Modality Embedding.** For multimodal input $U_m^1$, we employ an Embedding Encoder $\mathrm{E}_m^1$ with two Transformer encoder layers to extract and unify the feature. Each modality begins with a randomly initialized low-dimensional token $H_m^0 \in \mathbb{R}^{T \times d_m}$. These tokens are then processed by the Transformer encoder layer, embedding essential modality information and producing unified features, represented as $H_m^1 \in \mathbb{R}^{T \times d}$. The process is formalized by:

$$H_m^1 = \mathrm{E}_m^1 \left( \mathrm{concat} \left( H_m^0, U_m^1 \right) \right), \tag{1}$$

where $\mathrm{E}_m^1(\cdot)$ extracts features for each modality, and $\mathrm{concat}(\cdot)$ represents the concatenation operation.

**Adaptive Hyper-modality Learning.** In the original ALMT, each Adaptive Hyper-modality Learning layer contains a Transformer and two multi-head attention (MHA) modules. These are applied to learn language representations at different scales and hyper-modality representations from visual and audio modalities, guided by the language modality. Considering the possibility of severe interference in the language modality (*i.e.* dominant modality) due to random data missing, we designed a Dominant Modality Correction (DMC) module to generate the proxy dominant feature $\tilde{H}_p^1$ and construct corrected dominate feature $H_d^1$ (more details can be found in Section 3.4). Specifically, the process of learning corrected dominated representation $H_d^i$ at different scales can be described as:

$$H_d^i = \mathrm{E}_m^i \left( H_d^{i-1} \right), \tag{2}$$

where $i \in \{2, 3\}$ means the $i$-th layer of Adaptive Hyper-modality Learning module, $\mathrm{E}_m^i(\cdot)$ is the i-th Transformer encoder layer, $H_d^i \in \mathbb{R}^{T \times d}$ is corrected dominated feature at different scale. To learn the hyper-modality representation, the corrected dominated feature and audio/visual features are used to calculate Query and Key/Value, respectively. Briefly, the process can be written as follows:

$$H_{hyper}^i = H_{hyper}^{i-1} + \mathrm{MHA}(H_d^i, H_a^1) + \mathrm{MHA}(H_d^i, H_v^1), \tag{3}$$

where $\mathrm{MHA}(\cdot)$ represents multi-head attention, $H_{hyper}^i \in \mathbb{R}^{T \times d}$ is the hyper-modality feature. Note that the feature $H_{hyper}^0 \in \mathbb{R}^{T \times d}$ is a random initialized vector.

**Multimodal Fusion and Prediction.** With obtained $H_d^3$ and $H_{hyper}^3$, a Transformer encoder with a classifier at a depth of 4 layers is employed for multimodal fusion and sentiment prediction:

$$\hat{y} = \text{CrossTransformer}(H_d^3, H_{hyper}^3), \tag{4}$$

where $\hat{y}$ is the sentiment prediction.

### 3.4 Dominant Modality Correction

This module consists of two steps, *i.e.,* completeness check of dominant modality and proxy dominant feature generation using adversarial learning (Ganin and Lempitsky, 2015; Goodfellow et al., 2014).

**Completeness Check.** We apply an encoder $\text{E}_{cc}$ that consists of a Transformer encoder with a depth of two layers and a classifier for completeness check. For example, if the missing rate of dominate modality is 0.3, the label of completeness is 0.7. This completeness prediction $w$ can be obtained as follows:

$$w = \text{E}_{cc}\left(\text{Concat}\left(H_{cc}, H_l^1\right)\right), \tag{5}$$

where $H_{cc} \in \mathbb{R}^{T \times d}$ is a randomly initialized token for completeness prediction. We optimize this process using L2 loss:

$$\mathcal{L}_{cc} = \frac{1}{N_b} \sum_{k=0}^{N_b} \left\| w^k - \hat{w}^k \right\|_2^2, \tag{6}$$

where $N_b$ is the number of samples in the training set, $\hat{w}^k$ is the label of completeness of $k$-th sample.

**Proxy Dominant Feature Generation.** With the randomly initialized feature $H_p^0 \in \mathbb{R}^{T \times d}$, visual feature $H_v^1$ and audio feature $H_a^1$, we employ a Proxy Dominant Feature Generator $E_{DFG}$, which consists of two Transformer encoder layers. This setup generates the proxy dominant features $H_p^1 \in \mathbb{R}^{T \times d}$, designed to complement and correct the dominant modality. The corrected dominant feature $H_d^1 \in \mathbb{R}^{T \times d}$ is calculated by combining $H_p^1$ and the language feature $H_l^1$, weighted by the predicted completeness $w$:

$$H_p^1 = \text{E}_{DFG}\left(\text{Concat}\left(H_p^0, H_a^1, H_v^1\right), \theta_{DFG}\right), \tag{7}$$

$$H_d^1 = (1 - w) * H_p^1 + w * H_l^1, \tag{8}$$

where $\theta_{DFG}$ denotes the parameters of the Proxy Dominant Feature Generator $E_{DFG}$.

To ensure that the agent feature offers a distinct perspective from the visual and audio features, we utilize an effectiveness discriminator $D$. This discriminator includes a binary classifier and a Gradient Reverse Layer (GRL) (Ganin and Lempitsky, 2015) and is tasked with identifying the origin of the agent features:

$$\hat{y}_p = \text{D}\left(H_p^1 / H_l^1, \theta_D\right), \tag{9}$$

where $\theta_D$ represents the parameters of the effectiveness discriminator $D$, and $\hat{y}_p$ indicates the prediction of whether the input feature originates from the language modality.

In practice, the generator and the discriminator engage in an adversarial learning structure. The discriminator aims to identify whether the features are derived from the language modality, while the generator's objective is to challenge the discriminator's ability to make accurate predictions. This dynamic is encapsulated in the adversarial learning objective:

$$\min_{\theta_D} \max_{\theta_{DFG}} \mathcal{L}_{adv} = -\frac{1}{N_b} \sum_{k=0}^{N_b} y_p^k \cdot \log \hat{y}_p^k, \tag{10}$$

where $N_b$ is the number of samples in the training set, and $y_p^k$ indicates the label determining whether the input feature for the $k$-th sample originates from the visual or audio modality.

### 3.5 Reconstructor

Our experiments demonstrate that reconstructing missing information can significantly enhance regression metrics. More details about this are shown in Table 4. To address this, we have developed

a reconstructor, denoted as $E_{rec}$, which comprises two Transformer layers designed to effectively rebuild missing information of each modality. The operational equation for the reconstructor is:

$$\hat{U}_m^0 = \text{E}_m^{rec}\left(U_m^1\right) \tag{11}$$

where $\hat{U}_m^0$ is the reconstructed feature corresponding to the feature $U_m^0$.

To optimize the performance of the reconstructor, we apply an L2 loss function:

$$\mathcal{L}_{rec} = \frac{1}{N_b} \sum_{h=0}^{N_b} \sum_m \left\| U_m^{0\,k} - \hat{U}_m^{0\,k} \right\|_2^2, \tag{12}$$

where $U_m^{0\,k}$ and $\hat{U}_m^{0\,k}$ represent the original and reconstructed features with missing information for the $k$-th sample, respectively. This loss function helps minimize the discrepancies between the original and reconstructed features, thereby improving the accuracy of other components, such as Dominant Modality Correction and final sentiment prediction.

### 3.6 Overall Learning Objectives

To sum up, our method involves four learning objectives, including a completeness check loss $\mathcal{L}_{cc}$, an adversarial learning loss $\mathcal{L}_{adv}$ for proxy dominant feature generation, a reconstruction loss $\mathcal{L}_{rec}$ and one final sentiment prediction loss $\mathcal{L}_{sp}$. The sentiment prediction loss $\mathcal{L}_{sp}$ can be described as:

$$\mathcal{L}_{sp} = \frac{1}{N_b} \sum_{n=0}^{N_b} \|y^n - \hat{y}^n\|_2^2, \tag{13}$$

Therefore, the overall loss $\mathcal{L}$ can be written as:

$$\mathcal{L} = \alpha\mathcal{L}_{cc} + \beta\mathcal{L}_{adv} + \gamma\mathcal{L}_{rec} + \delta\mathcal{L}_{sp}, \tag{14}$$

where $\alpha$, $\beta$, $\gamma$ and $\delta$ are hyperparameters. On MOSI and MOSEI datasets, we empirically set them to 0.9, 0.8, 0.1, and 1.0, respectively. On the SIMS dataset, we empirically set them to 0.9, 0.6, 0.1, and 1.0, respectively.

## 4 Experiments and Analysis

In this section, we provide a comprehensive and fair comparison between the proposed LNLN and previous representative MSA methods on MOSI (Zadeh et al., 2016), MOSEI (Zadeh et al., 2018) and SIMS (Yu et al., 2020) datasets.

### 4.1 Datasets

**MOSI.** The dataset includes 2,199 multimodal samples, integrating visual, audio, and language modalities. It is divided into a training set of 1,284 samples, a validation set of 229 samples, and a test set of 686 samples. Every single sample has been given a sentiment score, varying from -3, indicating strongly negative sentiment, to 3, signifying strongly positive sentiment.

**MOSEI.** The dataset consists of 22,856 video clips sourced from YouTube. The sample is divided into 16,326 clips for training, 1,871 for validation, and 4,659 for testing. Each clip is labeled with a score, ranging from -3, denoting the strongly negative, to 3, denoting the strongly positive.

**SIMS.** The dataset is a Chinese multimodal sentiment dataset that includes 2,281 video clips sourced from different movies and TV series. It has been partitioned into 1,368 samples for training, 456 for validation, and 457 for testing. Each sample has been manually annotated with a sentiment score ranging from -1 (negative) to 1 (positive).

### 4.2 Evaluation Settings and Criteria

For a fair and comprehensive evaluation, we experiment ten times, setting the missing rates $r$ to predefined values from 0 to 0.9 with an increment of 0.1. For instance, 50% of the information is

randomly erased from each modality in the test data when $r = 0.5$. Unlike previous works (Yuan et al., 2021, 2024), we did not evaluate at $r = 1.0$, as this would imply complete data erasure from each modality, rendering the experiment non-informative. With the obtained results of each missing rate, we compute the average value as the model's overall performance under different levels of noise.

For evaluation criteria, we report the binary classification accuracy (Acc-2), the F1 score associated with Acc-2, and the mean absolute error (MAE). For Acc-2, we calculated accuracy and F1 in two ways: negative/positive (left-side value of /) and negative/non-negative (right-side value of /) on the MOSI and MOSEI datasets, respectively. Additionally, we provide the three-class accuracy (Acc-3), the seven-class accuracy (Acc-7), and the correlation of the model's prediction with humans (Corr) on the MOSI dataset. For the SIMS dataset, we report Acc-3, the five-class accuracy (Acc-5), and Corr. Due to the distinct focus of regression and classification metrics on different aspects of model performance, a model achieving the lowest error on regression metrics may not necessarily exhibit optimal performance on classification metrics. To comprehensively reflect the model capabilities, we select the best-performing checkpoint for each type of metric across all models in the comparisons, thus capturing the peak performance of both regression and classification aspects independently.

### 4.3 Implementation Details

We used PyTorch 2.2.1 to implement the method. The experiments were conducted on a PC with an AMD EPYC 7513 CPU and an NVIDIA Tesla A40. To ensure consistent and fair comparisons across all methods, we conducted each experiment three times using fixed random seeds of 1111, 1112, and 1113. Details of the hyperparameters are shown in Table 1. In addition, the result of MISA, Self-MM, MMIM, CENET, TETFN, and TFR-Net is reproduced by the authors from open source code in the MMSA[2] (Mao et al., 2022), which is a unified framework for MSA, using default hyperparameters. The result of ALMT is reproduced by the authors from open source code on Github[3].

Table 1: Hyperparameters of LNLN we use on the different datasets

|  | MOSI | MOSEI | SIMS |
|---|---|---|---|
| Vector Length $T$ | 8 | 8 | 8 |
| Vector Dimension $d$ | 128 | 128 | 128 |
| Batch Size | 64 | 64 | 64 |
| Initial Learning Rate | 1e-4 | 1e-4 | 1e-4 |
| Loss Weight $\alpha, \beta, \gamma, \delta$ | 0.9, 0.8, 0.1, 1 .0 | 0.9, 0.8, 0.1, 1.0 | 0.9, 0.6, 0.1, 1.0 |
| Optimizer | AdamW | AdamW | AdamW |
| Epochs | 200 | 200 | 200 |
| Warm Up | ✓ | ✓ | ✓ |
| Cosine Annealing | ✓ | ✓ | ✓ |
| Early Stop | ✓ | ✓ | ✓ |
| Seed | 1111,1112,1113 | 1111,1112,1113 | 1111,1112,1113 |

### 4.4 Robustness Comparison

Tables 2 and 3 show the robustness evaluation results on the MOSI, MOSEI, and SIMS datasets. As shown in Table 2, LNLN achieves state-of-the-art performance on most metrics. On the MOSI dataset, LNLN achieved a relative improvement of 9.46% on Acc-7 compared to the sub-optimal result obtained by MMIM, demonstrating the robustness of LNLN in the face of different noise effects. However, on the MOSEI dataset, LNLN achieves only sub-optimal performance on metrics such as Acc-7 and Acc-5. After analyzing the data distribution (see Appendix A.3) and the confusion matrix (see Appendix A.6), we believe that this is because LNLN does not overly bias the neutral category, which has a disproportionate number of samples in noisy scenarios. As shown in Table 3, LNLN achieves an improvement in F1 on the SIMS dataset, with a relative improvement of 9.17% on F1 compared to the sub-optimal result obtained by ALMT. Similar to CENET's performance on the MOSEI dataset, TETFN on the SIMS dataset has a tendency to predict inputs as weakly negative categories with high sample sizes, resulting in seemingly better performance on some metrics. Additionally, as shown in Figure 2, we present the performance curves of several advanced

---

[2]MMSA: `https://github.com/thuiar/MMSA`
[3]ALMT: `https://github.com/Haoyu-ha/ALMT`

methods under varying missing rates. The results demonstrate that the proposed LNLN consistently outperforms others across most scenarios, showing its robustness under different missing rates.

Overall, LNLN attempts to make predictions in the face of highly noisy inputs without the severe lazy behavior observed in other models, which often leads to predictions heavily biased towards a certain category. This demonstrates that LNLN shows strong robustness and competitiveness across various datasets and noise levels, highlighting its effectiveness in multimodal sentiment analysis.

Table 2: Robustness comparison of the overall performance on MOSI and MOSEI datasets. Note: The smaller MAE indicates the better performance.

| Method | MOSI | | | | | | MOSEI | | | | | |
|---|---|---|---|---|---|---|---|---|---|---|---|---|
| | Acc-7 | Acc-5 | Acc-2 | F1 | MAE | Corr | Acc-7 | Acc-5 | Acc-2 | F1 | MAE | Corr |
| MISA | 29.85 | 33.08 | 71.49 / 70.33 | 71.28 / 70.00 | 1.085 | 0.524 | 40.84 | 39.39 | 71.27 / 75.82 | 63.85 / 68.73 | 0.780 | 0.503 |
| Self-MM | 29.55 | 34.67 | 70.51 / 69.26 | 66.60 / 67.54 | 1.070 | 0.512 | 44.70 | 45.38 | 73.89 / 77.42 | 68.92 / 72.31 | 0.695 | 0.498 |
| MMIM | 31.30 | 33.77 | 69.14 / 67.06 | 66.65 / 64.04 | 1.077 | 0.507 | 40.75 | 41.74 | 73.32 / 75.89 | 68.72 / 70.32 | 0.739 | 0.489 |
| CENET | 30.38 | 37.25 | 71.46 / 67.73 | 68.41 / 64.85 | 1.080 | 0.504 | **47.18** | **47.83** | 74.67 / 77.34 | 70.68 / 74.08 | **0.685** | **0.535** |
| TETFN | 30.30 | 34.34 | 69.76 / 67.68 | 65.69 / 63.29 | 1.087 | 0.507 | 30.30 | 47.70 | 69.76 / 67.68 | 65.69 / 63.29 | 1.087 | 0.508 |
| TFR-Net | 29.54 | 34.67 | 68.15 / 66.35 | 61.73 / 60.06 | 1.200 | 0.459 | 46.83 | 34.67 | 73.62 / 77.23 | 68.80 / 71.99 | 0.697 | 0.489 |
| ALMT | 30.30 | 33.42 | 70.40 / 68.39 | 72.57 / **71.80** | 1.083 | 0.498 | 40.92 | 41.64 | **76.64** / 77.54 | 77.14 / 78.03 | 0.674 | 0.481 |
| **LNLN** | **34.26** | **38.27** | **72.55 / 70.94** | **72.73** / 71.25 | **1.046** | **0.527** | 45.42 | 46.17 | 76.30 / **78.19** | **77.77 / 79.95** | 0.692 | 0.530 |

Table 3: Robustness comparison of the overall performance on SIMS dataset. Note: The smaller MAE indicates the better performance.

| Method | Acc-5 | Acc-3 | Acc-2 | F1 | MAE | Corr |
|---|---|---|---|---|---|---|
| MISA | 31.53 | 56.87 | 72.71 | 66.30 | 0.539 | 0.348 |
| Self-MM | 32.28 | 56.75 | 72.81 | 68.43 | 0.508 | 0.376 |
| MMIM | 31.81 | 52.76 | 69.86 | 66.21 | 0.544 | 0.339 |
| CENET | 22.29 | 53.17 | 68.13 | 57.90 | 0.589 | 0.107 |
| TETFN | 33.42 | 56.91 | **73.58** | 68.67 | **0.505** | 0.387 |
| TFR-Net | 26.52 | 52.89 | 68.13 | 58.70 | 0.661 | 0.169 |
| ALMT | 20.00 | 45.36 | 69.66 | 72.76 | 0.561 | 0.364 |
| **LNLN** | **34.64** | **57.14** | 72.73 | **79.43** | 0.514 | **0.397** |

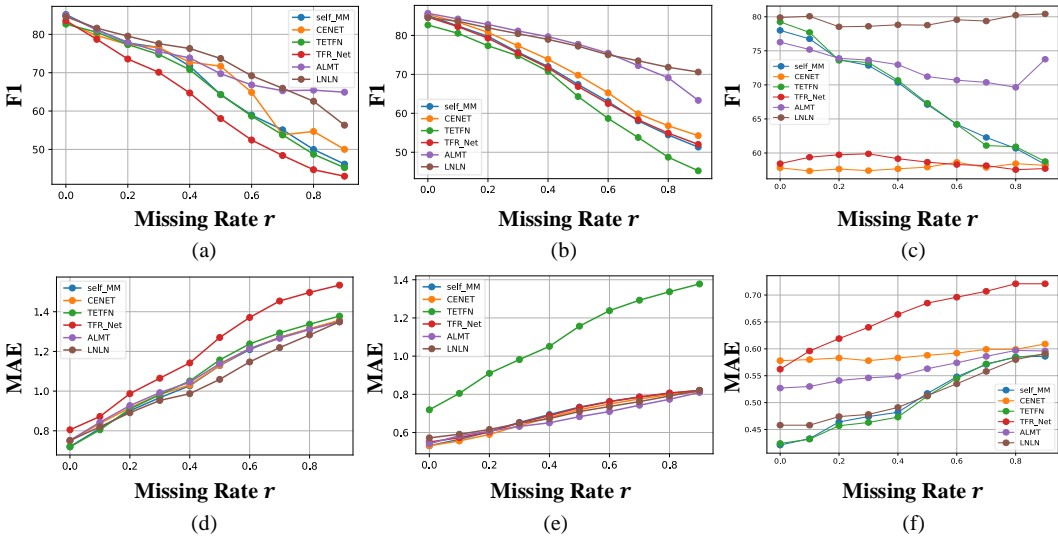

Figure 2: Performance curves of various missing rates. (a), (b) and (c) are the F1 curves on MOSI, MOSEI, and SIMS, respectively. (d), (e) and (f) are the MAE curves on MOSI, MOSEI, and SIMS, respectively. Note: The smaller MAE indicates the better performance.

## 4.5 Effects of Different Components

To verify the role of different components, we present the results after subtracting each component on the MOSI dataset separately. As shown in Table 4, some metrics show an upward trend in performance when certain modules are removed. On one hand, we believe this is due to randomness in noisy scenarios. On the other hand, the same hyperparameters cannot make the model perform optimally across all metrics. Moreover, the performance of LNLN decreases relatively slightly when the DMC and reconstructor are removed individually. However, there is a significant drop in performance when both are removed, *e.g.,* 4.72% decrease in Acc-7 and 5.87% decrease in Acc-5. Performance drops further after replacing DMML with concatenation fusion, which also proves that DMML plays an important role in MSA. Additionally, after removing the noisy data used for training, the performance of LNLN also shows a significant decrease. This observation indicates that LNLN needs to learn to be aware of noise with the help of noisy data. Overall, both the presence of noisy data and well-designed components are crucial for robust MSA in scenarios with random missing data.

Table 4: Effects of different components. Note: The smaller MAE indicates the better performance.

| Method | MOSI | | | | | | SIMS | | | | | |
|---|---|---|---|---|---|---|---|---|---|---|---|---|
| | Acc-7 | Acc-5 | Acc-2 | F1 | MAE | Corr | Acc-5 | Acc-3 | Acc-2 | F1 | MAE | Corr |
| **LNLN** | **34.26** | 38.27 | **72.55** / 70.94 | 72.73 / 71.25 | **1.046** | **0.527** | 34.64 | 57.14 | 72.73 | **79.43** | **0.514** | **0.397** |
| w/o DMC & Reconstructor | 29.54 | 32.40 | 71.98 / 70.31 | **73.53** / 71.29 | 1.104 | 0.478 | 33.81 | 54.00 | 72.57 | 78.10 | 0.525 | 0.376 |
| w/o DMC | 33.94 | 37.89 | 72.24 / 71.02 | 72.86 / 71.45 | 1.049 | 0.522 | 33.96 | **58.30** | **73.40** | 77.58 | 0.518 | 0.386 |
| w/o Reconstructor | 33.94 | **38.52** | 70.81 / **71.35** | 71.17 / **71.69** | 1.054 | 0.518 | 34.02 | 58.29 | 73.07 | 78.71 | 0.548 | 0.351 |
| w/o DMML | 33.22 | 37.25 | 69.55 / 69.55 | 70.81 / 70.71 | 1.084 | 0.504 | **34.87** | 56.06 | 70.74 | 76.94 | 0.553 | 0.245 |
| w/o Missing Data for Training | 30.11 | 34.82 | 70.62 / 68.78 | 72.49 / 71.03 | 1.105 | 0.486 | 33.48 | 53.09 | 72.96 | 79.17 | 0.519 | 0.360 |

## 4.6 Effects of Different Regularization

As shown in Table 5, we present the ablation results after subtracting each regularization on the MOSI dataset separately. Obviously, after each regularization is removed, LNLN shows a decrease in performance on most metrics. This demonstrates that regularization contributes to the learning of the model. Similar to the results in Table 4, there are a few metrics that improve after the removal of regularization. We believe this is mainly due to the difficulty of optimizing a single set of hyperparameters to achieve the best performance across all metrics.

Table 5: Effects of different regularization. Note: The smaller MAE indicate the better performance.

| Method | MOSI | | | | | | SIMS | | | | | |
|---|---|---|---|---|---|---|---|---|---|---|---|---|
| | Acc-7 | Acc-5 | Acc-2 | F1 | MAE | Corr | Acc-5 | Acc-3 | Acc-2 | F1 | MAE | Corr |
| **LNLN** | **34.26** | 38.27 | **72.55** / **70.94** | 72.73 / **71.25** | **1.046** | 0.527 | **34.64** | **57.14** | 72.73 | **79.43** | **0.514** | **0.397** |
| w/o $\mathcal{L}_{cc}$ | 33.61 | 37.78 | 72.32 / 70.72 | 72.55 / 71.02 | 1.047 | 0.531 | 32.87 | 56.41 | 70.78 | 75.30 | 0.523 | 0.376 |
| w/o $\mathcal{L}_{adv}$ | 32.84 | 37.79 | 72.28 / 70.31 | **72.83** / 70.40 | 1.052 | 0.526 | 32.47 | 54.24 | **73.80** | 73.91 | 0.525 | 0.351 |
| w/o $\mathcal{L}_{rec}$ | 33.94 | **38.52** | 72.46 / 70.81 | 72.69 / 71.17 | 1.048 | 0.523 | 34.33 | 56.12 | 71.29 | 74.73 | 0.525 | 0.364 |

## 4.7 Effects of Different Modalities

To verify the role of different modalities, we present the ablation results after subtracting each modality from the MOSI dataset. As shown in Table 6, all modalities contribute to the final performance. Notably, when the language modality is removed, the performance of LNLN shows a significant decrease, while removing the visual and audio modalities results in a smaller decrease. This indicates that the language modality plays a crucial role in the MSA dataset. Moreover, we observe that LNLN converges on most metrics even when the visual and audio modalities are removed. This is due to the model's lazy behavior, where LNLN tends to fix its predictions to a certain category. More details can be found in Appendix A.3 and A.6.

## 4.8 Discussion

In this work, we conduct a comprehensive evaluation of current advanced methods, the results of which have triggered some thoughts. Specifically, we find that all methods fail to accurately predict

Table 6: Effects of different modalities. Note: The smaller MAE indicates the better performance.

| Method | MOSI | | | | | |
| --- | --- | --- | --- | --- | --- | --- |
| | Acc-7 | Acc-5 | Acc-2 | F1 | MAE | Corr |
| **LNLN** | **34.26** | **38.27** | **72.55 / 70.94** | 72.73 / **71.25** | **1.046** | 0.527 |
| w/o language | 16.30 | 16.31 | 54.88 / 55.79 | 60.33 / 63.10 | 1.399 | 0.033 |
| w/o audio | 33.51 | 38.10 | 72.31 / 70.54 | 73.67 / 71.10 | 1.064 | 0.535 |
| w/o vision | 33.53 | 38.15 | 72.32 / 70.58 | **73.68** / 71.14 | 1.064 | **0.536** |

the inputs in high missing rate scenarios because most of them are designed specifically for complete data (see Appendix A.6 for more details). However, some methods show relatively more robust performance in low missing rate scenarios and some specific scenarios (see Appendix A.8), such as Self-MM, TETFN, and ALMT. We believe that some of the ideas in these methods may be useful for future research on more robust MSA for incomplete data.

For example, the Unimodal Label Generation Module (ULGM) (Yu et al., 2021) used in both Self-MM and TETFN facilitates the modeling of incomplete data. Due to the random data missing, the temporal and structured affective information in the data is corrupted, making it difficult for the models to perceive consistency and variability information between different modalities. Using ULGM to generate pseudo-labels as an additional supervisory signal for the models may directly enable the model to learn the relationship between different modalities. For ALMT, we believe that its Adaptive Hyper-modality Learning (AHL) module facilitates the modeling of robust affective representations. AHL mainly uses the dominant modality to compute query vectors and applies multi-head attention to query useful sentiment cues from other modalities as a complement to the dominant modality. This process may reduce the difficulty of multimodal fusion by avoiding the introduction of sentiment-irrelevant information to some extent. Our proposed LNLN, based on this idea from ALMT, proves the effectiveness of the method.

Additionally, we explore the performance of these methods in modality missing scenarios, which is a special case of random data missing where the partial modality missing rate $r = 1.0$. We find that when language modality is missing, the performance of many models decreases significantly and converges to the same value. More details can be seen in Appendix A.8.

## 5 Conclusion and Future Work

In this paper, a novel Language-dominated Noise-resistant Learning Network (LNLN) is proposed for robust MSA. Due to the collaboration between Dominate Modality Correction (DMC) module, the Dominant Modality based Multimodal Learning (DMML) module, and the Reconstructor, LNLN can enhance the dominant modality to achieve superior performance in data missing scenarios with varying levels. Extensive evaluation shows that none of the existing methods can effectively model the data with high missing rates. The related analyses can provide suggestions for other researchers to better handle robust MSA. In the future, we will focus on improving the generalization of the model to handle different types of scenes and varying intensities of noise effectively.

## Limitations

We believe there are several limitations to the LNLN. 1) The LNLN achieves good performance in data missing scenarios, but is not always better than all other methods in the modality missing scenarios, which demonstrates the lack of multi-scene generalization capability of the LNLN. 2) The data in real-world scenarios is much more complex. In addition to the presence of missing data, other factors need to be considered, such as diverse cultural contexts, varying user behavior patterns, and the influence of platform-specific features on the data. These factors can introduce additional noise and variability, which may require further model adaptation and tuning to handle effectively. 3) Tuning the hyperparameters, particularly those related to the loss functions, can be challenging and may require more sophisticated methods to achieve optimal performance.

## Acknowledgements

This work is supported by the Guangdong Provincial Key Laboratory of Mathematical Foundations for Artificial Intelligence (2023B1212010001).

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

# A Additional Experiments and Analysis

## A.1 Effect of Regularization Weight on Model Performance

As shown in Table 7, we empirically tried different combinations of hyperparameters in the loss function. The results show that the selected parameters can make the LNLN achieve better performance in most metrics. This also demonstrates the effectiveness of the optimization objectives of LNLN.

Table 7: Effect of regularization weight on model performance. Note: The smaller MAE indicate the better performance.

| MOSI | | | | | | |
|---|---|---|---|---|---|---|
| $\alpha, \beta, \gamma, \delta$ | Acc-7 | Acc-5 | Acc-2 | F1 | MAE | Corr |
| 0.9, 0.8, 0.1, 1.0 | **34.26** | 38.27 | **72.55** / 70.94 | 72.73 / 71.25 | **1.046** | **0.527** |
| 0.9, 0.8, 0, 1.0 | 33.94 | **38.52** | 72.46 / 70.81 | 72.69 / 71.17 | 1.148 | 0.523 |
| 0.9, 0.8, 0.2, 1.0 | 32.76 | 37.33 | 72.20 / 70.69 | 72.43 / 71.22 | 1.078 | 0.511 |
| 0.9, 0, 0.1, 1.0 | 32.84 | 37.79 | 72.28 / 70.31 | 72.83 / 70.40 | 1.052 | 0.526 |
| 0.9, 0.5, 0.1, 1.0 | 32.43 | 37.23 | 72.04 / 70.24 | 72.59 / 70.96 | 1.063 | 0.511 |
| 0.9, 1.0, 0.1, 1.0 | 32.24 | 37.07 | 71.61 / 70.72 | **73.04** / 70.89 | 1.063 | 0.518 |
| 0, 0.8, 0.1, 1.0 | 33.61 | 37.78 | 72.32 / 70.72 | 72.55 / 71.02 | 1.047 | 0.531 |
| 0.5, 0.8, 0.1, 1.0 | 33.29 | 37.78 | 72.40 / **70.95** | 72.61 / 71.29 | 1.067 | 0.515 |
| 1.0, 0.8, 0.1, 1.0 | 33.22 | 36.64 | 71.66 / 70.57 | 72.94 / **71.80** | 1.067 | 0.511 |
| SIMS | | | | | | |
| $\alpha, \beta, \gamma, \delta$ | Acc-5 | Acc-3 | Acc-2 | F1 | MAE | Corr |
| 0.9, 0.6, 0.1, 1.0 | 34.64 | **57.14** | 72.73 | **79.43** | **0.514** | **0.397** |
| 0.9, 0.6, 0, 1.0 | 32.87 | 56.41 | 70.78 | 75.30 | 0.523 | 0.376 |
| 0.9, 0.6, 0.2, 1.0 | 33.95 | 55.11 | 70.82 | 73.20 | 0.517 | 0.390 |
| 0.9, 0, 0.1, 1.0 | 32.47 | 54.24 | **73.80** | 73.91 | 0.525 | 0.351 |
| 0.9, 0.3, 0.1, 1.0 | **35.16** | 56.57 | 72.86 | 77.54 | 0.524 | 0.358 |
| 0.9, 1.0, 0.1, 1.0 | 32.92 | 54.66 | 72.50 | 76.94 | 0.543 | 0.363 |
| 0, 0.6, 0.1, 1.0 | 34.33 | 56.12 | 71.29 | 74.73 | 0.525 | 0.364 |
| 0.5, 0.6, 0.1, 1.0 | 33.96 | 55.57 | 70.63 | 73.06 | 0.539 | 0.360 |
| 1.0, 0.6, 0.1, 1.0 | 33.25 | 57.03 | 70.57 | 72.20 | 0.522 | 0.367 |

## A.2 Analysis of Model Stability

To evaluate the stability of the model, we selected the MOSI dataset for ablation experiments. Specifically, based on the overall performance (mean) in Table 2, we selected several representative methods to additionally compute the standard deviation. For each method, we first calculate the standard deviation of evaluation results for the three random seeds when the missing rate $r$ ranges from 0 to 0.9, and then average the 10 sets of standard deviation. The final result is shown in Table 8. We can see that the standard deviation of these methods is not very large in most metrics, indicating that all these methods can guarantee stability in the presence of random noise. It should be noted that LNLN strikes a balance between overall performance and stability.

## A.3 Analysis of Data Distribution

Figure 4 illustrates the data distribution on the MOSI, MOSEI, and SIMS datasets. Significant category imbalances can be seen across all datasets. Additionally, the distributions of the training set, validation set, and test set are not identical on the MOSI and MOSEI datasets. For example, as shown in Figure 4 (a), the percentage of weakly negative samples is 15.5% in the MOSI training set, while this category reaches 22.7% in the test set, representing the highest percentage among all categories.

Table 8: Comparison of model stability on MOSI dataset. Note: -±- means mean±std. The smaller MAE indicates better performance.

| Method | Acc-7 | Acc-5 | Acc-2 | F1 | MAE | Corr |
|--------|-------|-------|-------|-----|-----|------|
| MISA | 29.85±2.62 | 33.08±1.77 | 71.49±1.00 / 70.33±0.93 | 71.28±0.92/ 70.00±1.26 | 1.085±0.08 | 0.524±0.08 |
| Self-MM | 29.55±1.06 | 34.67±1.87 | 70.51±0.71 / 69.26±1.07 | 66.60±1.92 / 67.54±2.34 | 1.070±0.13 | 0.512±0.07 |
| CENET | 30.38±1.27 | 37.25±1.53 | 71.46±0.60 / 67.73±0.71 | 68.41±1.12 / 64.85±2.56 | 1.080±0.14 | 0.504±0.11 |
| TFR-Net | 29.54±1.00 | 34.67±1.75 | 68.15±1.25 / 66.35±1.09 | 61.73±2.82 / 60.06±2.33 | 1.200±0.11 | 0.459±0.25 |
| **LNLN** | **34.26**±1.17 | **38.27**±1.23 | **72.55**±1.11 / **70.94**±1.28 | **72.73**±0.99 / **71.25**±1.06 | **1.046**±0.21 | **0.527**±0.17 |

Similarly, Figure 4 (b) and Figure 4 (c) show that this phenomenon also exists in the MOSEI and SIMS datasets.

The unbalanced data distribution makes it difficult for the model to perform well in data missing scenarios. Specifically, it is easy for the model to engage in lazy behavior in high-noise scenarios, simply biasing predictions towards categories with a higher proportion in the training set. These unbalanced distributions may further increase the learning difficulty of the model when facing data missing. More details can be found in Appendix A.6.

## A.4 Case Study

As shown in Figure 3, we visualize several successful and failed predictions made by LNLN and ALMT from the MOSI dataset for the case study. It shows that LNLN can perceive sentiment cues in challenging samples, which demonstrates its ability to capture sentiment cues in noisy scenarios. However, for inputs with high missing rates (*e.g.,* the third example in the figure), both LNLN and ALMT fail to make correct predictions. We believe this is due to the high loss of valid information in the multimodal input, making accurate predictions difficult.

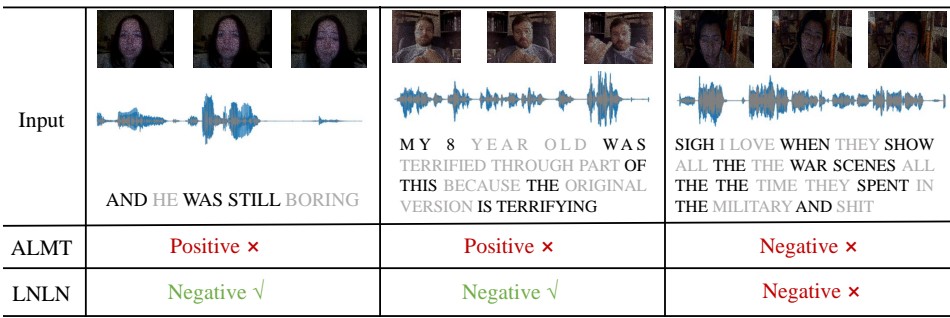

Figure 3: Visualization of successful and failed predictions. Note: The input is visualized to facilitate readers' understanding. In practice, random data missing is applied to the original input sequence, as described in Section 3.

## A.5 Details of Robust Comparison

Tables 9 and 10 show the details of robust comparison on the MOSI, MOSEI, and SIMS datasets, respectively. We observe that Self-MM and TETFN have a clear advantage in many evaluation metrics when the missing rate $r$ is low. As the missing rate $r$ increases, LNLN shows a significant improvement in all evaluation metrics. This demonstrates LNLN's ability to effectively model data affected by noise of varying intensity. In general, models augmented with noisy data in training usually cannot achieve state-of-the-art performance when the noise is low due to differences in data distribution. It is also worthwhile to study in the future how to balance the performance of models under different levels of noise.

In addition, when $r$ is high, many models converge in their performance on metrics such as Acc-7 and Acc-5 (see Appendix A.6). This is due to the fact that these models exhibit lazy behavior, tending to predict inputs as categories with a higher number of samples in the training set. In such cases, we

believe that the models do not actually learn effective knowledge, even if their performance appears good.

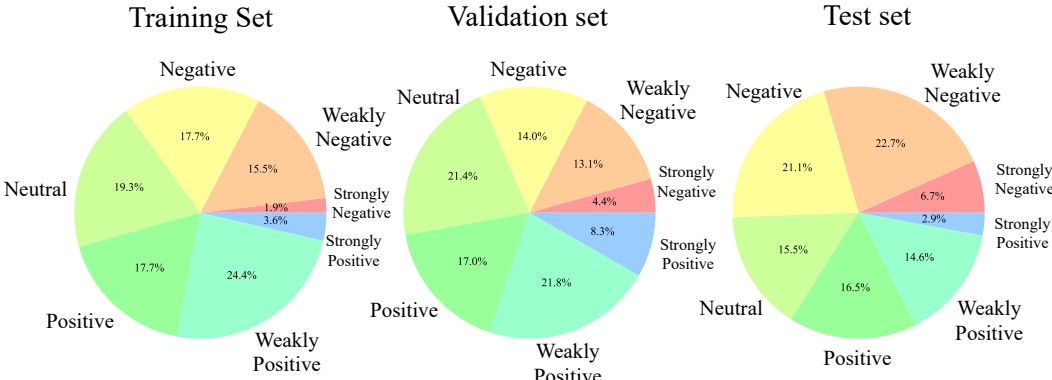

(a) Data distribution of MOSI dataset

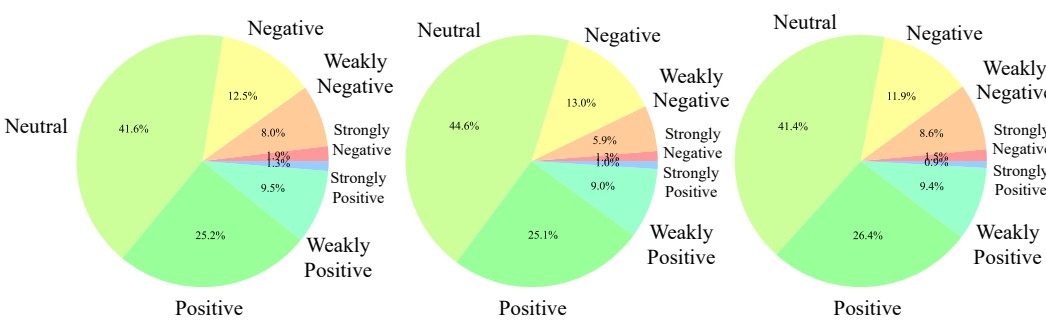

(b) Data distribution of MOSEI dataset

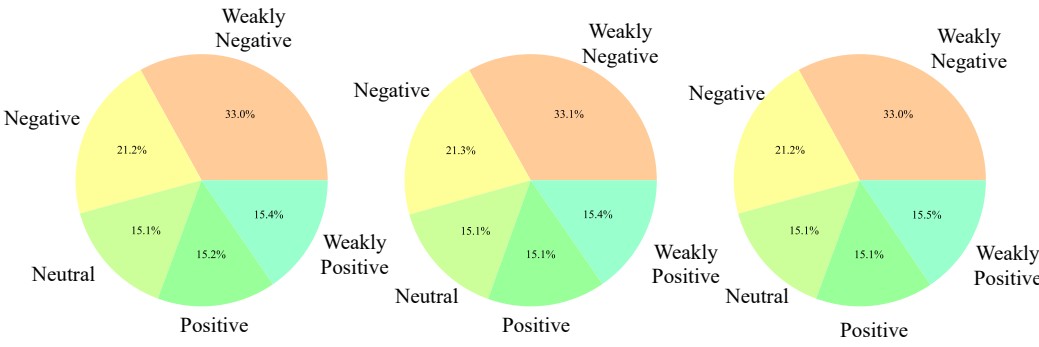

(c) Data distribution of SIMS dataset

Figure 4: Data distribution of MOSI, MOSEI, and SIMS datasets.

## A.6 Visualization of Confusion Matrix

To verify the effectiveness of the method, we visualized the confusion matrix of several representative methods on the MOSI, MOSEI, and SIMS datasets in Figure 5, Figure 6, and Figure 7, respectively. Obviously, as the missing rate $r$ increases, the predictions of all methods on all datasets tend to favor certain categories. This phenomenon indicates that the models hardly learn useful knowledge for prediction but instead ensure high accuracy by being lazy. For example, on the MOSI dataset,

Table 9: Details of robust comparison on MOSI and MOSEI with different random missing rates. Note: The smaller MAE indicates the better performance.

| Method | MOSI | | | | | | MOSEI | | | | | |
|---|---|---|---|---|---|---|---|---|---|---|---|---|
| | Acc-7 | Acc-5 | Acc-2 | F1 | MAE | Corr | Acc-7 | Acc-5 | Acc-2 | F1 | MAE | Corr |
| Random Missing Rate $r = 0$ | | | | | | | | | | | | |
| MISA | 43.05 | 48.30 | 82.78 / 81.24 | 82.83 / 81.23 | 0.771 | 0.777 | 51.79 | 53.85 | 85.28 / 84.10 | 85.10 / 83.75 | 0.552 | 0.759 |
| Self-MM | 42.81 | **52.38** | **85.22 / 83.24** | **85.19 / 83.26** | 0.720 | 0.790 | 53.89 | 55.72 | 85.34 / **84.68** | 85.11 / **84.66** | **0.531** | 0.764 |
| MMIM | **45.92** | 49.85 | 83.43 / 81.97 | 83.43 / 81.94 | 0.744 | 0.778 | 50.76 | 53.04 | 83.53 / 81.65 | 83.39 / 81.41 | 0.576 | 0.724 |
| CENET | 43.20 | 50.39 | 83.08 / 81.49 | 83.06 / 81.48 | 0.748 | 0.785 | **54.39** | **56.12** | 85.49 / 82.30 | 85.41 / 82.60 | **0.531** | 0.770 |
| TETFN | 44.07 | 51.31 | 82.62 / 81.10 | 82.67 / 81.09 | **0.719** | **0.794** | 44.07 | 55.96 | 82.62 / 81.10 | 82.67 / 81.09 | 0.719 | **0.794** |
| TFR-Net | 40.82 | 47.91 | 83.64 / 81.68 | 83.57 / 81.61 | 0.805 | 0.760 | 53.71 | 47.91 | 84.96 / 84.65 | 84.71 / 84.34 | 0.550 | 0.745 |
| ALMT | 42.37 | 48.49 | 84.91 / 82.75 | 85.01 / 82.94 | 0.752 | 0.768 | 52.18 | 53.89 | **85.62 / 83.99** | **85.69 / 84.53** | 0.542 | 0.752 |
| LNLN | 44.56 | 49.76 | 84.25 / 81.24 | 84.61 / 81.79 | 0.751 | 0.778 | 50.66 | 51.94 | 84.14 / 83.61 | 84.53 / 84.02 | 0.572 | 0.735 |
| Random Missing Rate $r = 0.1$ | | | | | | | | | | | | |
| MISA | 40.28 | 46.21 | 80.18 / 79.01 | 80.21 / 78.97 | 0.847 | 0.721 | 50.13 | 51.34 | 82.21 / 82.28 | 81.28 / 80.79 | 0.598 | 0.722 |
| Self-MM | 40.33 | **49.03** | **81.40 / 80.03** | 81.19 / **80.03** | 0.812 | 0.728 | 51.80 | 53.18 | 83.03 / **83.79** | 82.43 / 83.23 | 0.564 | 0.725 |
| MMIM | 42.61 | 46.65 | 79.98 / 78.13 | 79.83 / 77.99 | 0.825 | 0.718 | 49.09 | 51.19 | 82.00 / 81.09 | 81.57 / 80.15 | 0.602 | 0.696 |
| CENET | 40.13 | 46.60 | 80.08 / 78.38 | 79.91 / 78.20 | 0.837 | 0.719 | **52.83** | 54.23 | 83.75 / 82.41 | 83.42 / 82.34 | **0.556** | **0.739** |
| TETFN | 40.67 | 46.84 | 80.59 / 78.91 | 80.55 / 78.79 | **0.805** | **0.731** | 40.67 | **54.28** | 80.59 / 78.91 | 80.55 / 78.79 | 0.805 | 0.731 |
| TFR-Net | 38.63 | 45.82 | 79.27 / 77.99 | 78.70 / 77.61 | 0.872 | 0.705 | 52.29 | 45.82 | 82.92 / 83.31 | 82.25 / 82.40 | 0.573 | 0.715 |
| ALMT | 39.84 | 45.48 | 80.90 / 78.67 | 81.15 / 79.08 | 0.843 | 0.703 | 49.98 | 51.38 | **84.14 / 82.84** | **84.23 / 83.04** | 0.583 | 0.718 |
| LNLN | **42.37** | 47.91 | 81.20 / 78.43 | **81.62** / 79.04 | 0.820 | 0.724 | 51.25 | 51.25 | 83.32 / 82.73 | 83.66 / 82.91 | 0.591 | 0.712 |
| Random Missing Rate $r = 0.2$ | | | | | | | | | | | | |
| MISA | 36.25 | 41.55 | 77.54 / 76.34 | 77.58 / 76.30 | 0.939 | 0.654 | 47.24 | 47.66 | 77.84 / 79.93 | 75.56 / 76.88 | 0.659 | 0.674 |
| Self-MM | 36.64 | 43.98 | 78.15 / 76.48 | 77.76 / 76.51 | 0.901 | 0.660 | 49.44 | 50.51 | 80.84 / **82.33** | 79.76 / 81.17 | 0.604 | 0.678 |
| MMIM | 39.07 | 42.66 | 76.42 / 74.54 | 76.12 / 74.22 | 0.918 | 0.651 | 46.27 | 47.99 | 79.93 / 79.66 | 79.08 / 77.68 | 0.642 | 0.653 |
| CENET | 38.00 | 42.32 | 77.49 / 74.64 | 77.35 / 74.28 | 0.916 | 0.654 | **50.72** | 51.85 | 81.46 / 81.62 | 80.78 / 81.17 | **0.590** | **0.698** |
| TETFN | 35.81 | 41.79 | 77.49 / 75.60 | 77.35 / 75.35 | 0.910 | 0.657 | 35.81 | **52.35** | 77.49 / 75.60 | 77.35 / 75.35 | 0.910 | 0.657 |
| TFR-Net | 34.70 | 40.13 | 74.70 / 73.52 | 73.57 / 72.70 | 0.987 | 0.622 | 51.04 | 40.13 | 80.87 / 81.61 | 79.29 / 79.99 | 0.604 | 0.672 |
| ALMT | 35.33 | 40.33 | 77.64 / 75.70 | 77.94 / 76.24 | 0.927 | 0.645 | 46.61 | 47.82 | **82.71** / 81.65 | **82.82** / 81.83 | 0.607 | 0.669 |
| LNLN | **39.74** | **45.14** | **79.22 / 76.87** | **79.53 / 77.34** | **0.891** | **0.668** | 48.75 | 49.95 | 81.70 / 81.68 | 81.95 / **81.89** | 0.616 | 0.677 |
| Random Missing Rate $r = 0.3$ | | | | | | | | | | | | |
| MISA | 34.60 | 38.97 | 75.76 / 74.54 | 75.82 / 74.51 | 0.989 | **0.618** | 43.99 | 43.40 | 73.32 / 77.28 | 68.91 / 72.25 | 0.724 | 0.615 |
| Self-MM | 34.89 | 40.67 | 76.37 / 74.98 | 75.68 / 74.94 | 0.967 | 0.614 | 47.23 | 48.07 | 77.63 / 79.99 | 75.69 / 77.74 | 0.653 | 0.610 |
| MMIM | 36.83 | 40.43 | 74.08 / 71.91 | 73.47 / 71.28 | 0.974 | 0.612 | 43.25 | 44.73 | 77.08 / 77.79 | 75.46 / 74.49 | 0.690 | 0.597 |
| CENET | 34.74 | 38.97 | 76.83 / 72.01 | 76.56 / 71.30 | 0.983 | 0.605 | 48.49 | 49.37 | 78.65 / 80.02 | 77.34 / 78.94 | 0.636 | **0.640** |
| TETFN | 33.24 | 38.58 | 75.25 / 73.42 | 74.77 / 72.78 | 0.982 | 0.607 | 33.24 | **50.23** | 75.25 / 73.42 | 74.77 / 72.78 | 0.982 | 0.607 |
| TFR-Net | 32.55 | 38.34 | 72.36 / 71.28 | 70.12 / 69.58 | 1.065 | 0.572 | **48.75** | 38.34 | 77.48 / 79.29 | 75.43 / 76.52 | 0.650 | 0.604 |
| ALMT | 33.04 | 37.17 | 75.15 / 72.94 | 75.51 / 73.66 | 0.992 | 0.596 | 43.04 | 44.05 | **80.94** / 79.94 | **81.15** / 80.20 | **0.632** | 0.598 |
| LNLN | **38.00** | **42.81** | **77.29 / 75.46** | **77.56 / 75.68** | **0.953** | 0.617 | 47.36 | 48.40 | 80.11 / **80.45** | 80.44 / **80.91** | 0.648 | 0.629 |
| Random Missing Rate $r = 0.4$ | | | | | | | | | | | | |
| MISA | 32.65 | 35.37 | 73.88 / 72.59 | 73.88 / 72.49 | 1.041 | 0.585 | 40.87 | 39.53 | 70.46 / 75.04 | 64.02 / 67.93 | 0.780 | 0.561 |
| Self-MM | 31.20 | 36.30 | 73.17 / 71.96 | 71.74 / 71.75 | 1.027 | 0.579 | 44.40 | 45.04 | 75.02 / 78.09 | 72.01 / 74.48 | 0.694 | 0.554 |
| MMIM | 33.38 | 35.76 | 70.84 / 68.90 | 69.69 / 67.80 | 1.034 | 0.576 | 40.84 | 41.86 | 74.56 / 76.15 | 71.98 / 71.40 | 0.732 | 0.542 |
| CENET | 32.26 | 36.15 | 73.38 / 71.53 | 72.75 / 70.26 | 1.031 | 0.574 | **47.12** | **47.74** | 76.03 / 78.57 | 73.87 / 76.75 | 0.678 | 0.587 |
| TETFN | 30.66 | 35.48 | 72.05 / 70.07 | 70.79 / 68.58 | 1.051 | 0.571 | 30.66 | 47.62 | 72.05 / 70.07 | 70.79 / 68.58 | 1.051 | 0.571 |
| TFR-Net | 30.17 | 35.76 | 68.75 / 67.74 | 64.71 / 64.41 | 1.142 | 0.537 | 46.70 | 35.76 | 74.74 / 77.65 | 71.67 / 73.71 | 0.688 | 0.548 |
| ALMT | 31.44 | 35.03 | 73.12 / 71.14 | 73.85 / 72.47 | 1.035 | 0.560 | 40.40 | 41.21 | **79.40** / 79.16 | **79.68** / 79.50 | **0.651** | 0.536 |
| LNLN | **36.49** | **41.11** | **76.01 / 74.25** | **76.31 / 74.67** | **0.987** | **0.594** | 45.99 | 46.88 | 78.49 / **79.70** | 78.98 / **80.46** | 0.673 | **0.592** |
| Random Missing Rate $r = 0.5$ | | | | | | | | | | | | |
| MISA | 28.14 | 30.61 | 70.53 / 69.34 | 70.50 / 69.20 | 1.124 | 0.519 | 38.12 | 36.05 | 67.38 / 73.21 | 58.38 / 64.14 | 0.834 | 0.492 |
| Self-MM | 26.97 | 31.39 | 67.43 / 67.54 | 64.27 / 66.81 | 1.129 | 0.503 | 42.70 | 43.14 | 71.97 / 75.81 | 67.40 / 70.38 | 0.733 | 0.477 |
| MMIM | 28.23 | 29.89 | 68.09 / 66.52 | 66.15 / 64.59 | 1.128 | 0.501 | 38.68 | 39.21 | 71.75 / 74.45 | 67.70 / 67.96 | 0.775 | 0.470 |
| CENET | 28.33 | 30.90 | 72.46 / 66.08 | 71.10 / 63.50 | 1.130 | 0.496 | **45.12** | 44.03 | 73.33 / 77.16 | 69.80 / 74.14 | 0.720 | 0.515 |
| TETFN | 27.55 | 31.34 | 67.23 / 65.06 | 64.30 / 61.78 | 1.157 | 0.492 | 27.55 | **45.63** | 67.23 / 65.06 | 64.30 / 61.78 | 1.157 | 0.492 |
| TFR-Net | 25.85 | 30.71 | 64.83 / 63.02 | 58.04 / 56.64 | 1.270 | 0.443 | 45.00 | 30.71 | 71.53 / 75.69 | 66.88 / 70.07 | 0.730 | 0.471 |
| ALMT | 28.42 | 31.25 | 68.24 / 65.94 | 69.74 / 68.54 | 1.138 | 0.485 | 37.82 | 38.34 | **77.40** / 77.48 | **77.73** / 77.80 | **0.683** | 0.461 |
| LNLN | **33.92** | **38.39** | **73.37 / 71.86** | **73.70 / 72.30** | **1.059** | **0.536** | 44.90 | 45.59 | 76.44 / **78.10** | 77.23 / **79.30** | 0.710 | **0.529** |
| Random Missing Rate $r = 0.6$ | | | | | | | | | | | | |
| MISA | 24.68 | 27.12 | 66.97 / 65.84 | 66.94 / 65.69 | 1.200 | 0.441 | 36.16 | 33.30 | 65.55 / 72.30 | 54.64 / 62.12 | 0.875 | 0.415 |
| Self-MM | 24.34 | 27.31 | 63.47 / 63.36 | 58.94 / 62.07 | 1.209 | 0.425 | 41.47 | 41.75 | 69.33 / 73.93 | 63.01 / 66.76 | 0.762 | 0.401 |
| MMIM | 25.41 | 27.11 | 63.67 / 62.49 | 60.87 / 59.48 | 1.208 | 0.418 | 37.13 | 37.48 | 68.83 / 73.16 | 63.09 / 65.43 | 0.808 | 0.402 |
| CENET | 24.54 | 26.53 | 67.58 / 61.47 | 64.87 / 57.86 | 1.215 | 0.415 | **44.45** | **44.64** | 70.50 / 75.39 | 65.27 / 70.86 | 0.749 | 0.446 |
| TETFN | 25.12 | 27.99 | 63.42 / 61.23 | 58.68 / 56.08 | 1.238 | 0.417 | 25.12 | 44.03 | 63.42 / 61.23 | 58.68 / 56.08 | 1.238 | 0.417 |
| TFR-Net | 24.05 | 28.33 | 61.64 / 59.47 | 52.44 / 50.53 | 1.371 | 0.363 | 43.88 | 28.33 | 68.80 / 74.05 | 62.51 / 67.07 | 0.762 | 0.397 |
| ALMT | 25.41 | 27.36 | 64.53 / 62.15 | 66.81 / 65.87 | 1.214 | 0.407 | 35.99 | 36.30 | **74.98** / 76.26 | **75.44** / 76.71 | **0.710** | 0.395 |
| LNLN | **30.37** | **34.35** | **69.00 / 67.69** | **69.19 / 67.99** | **1.147** | **0.458** | 43.52 | 44.00 | 73.82 / **76.50** | 75.03 / **78.33** | 0.736 | **0.471** |
| Random Missing Rate $r = 0.7$ | | | | | | | | | | | | |
| MISA | 21.14 | 23.27 | 65.09 / 63.89 | 65.07 / 63.74 | 1.257 | 0.381 | 34.54 | 31.21 | 64.28 / 71.71 | 51.82 / 60.65 | 0.906 | 0.344 |
| Self-MM | 20.70 | 23.81 | 61.74 / 61.46 | 55.11 / 58.97 | 1.271 | 0.339 | 39.93 | 40.12 | 66.79 / 72.55 | 58.05 / 63.45 | 0.786 | 0.329 |
| MMIM | 22.35 | 24.00 | 61.23 / 59.18 | 57.15 / 54.36 | 1.267 | 0.342 | 35.25 | 35.47 | 66.89 / 72.26 | 58.90 / 63.26 | 0.834 | 0.341 |
| CENET | 22.35 | 25.12 | 63.82 / 59.43 | 53.79 / 54.22 | 1.269 | 0.335 | **43.93** | **44.03** | 67.50 / 73.39 | 59.88 / 67.02 | 0.776 | 0.384 |
| TETFN | 23.13 | 25.27 | 61.13 / 58.65 | 53.79 / 50.77 | 1.293 | 0.337 | 23.13 | 43.15 | 61.13 / 58.65 | 53.79 / 50.77 | 1.293 | 0.337 |
| TFR-Net | 23.71 | 26.92 | 59.91 / 57.34 | 48.41 / 45.48 | 1.454 | 0.276 | 42.91 | 26.92 | 66.64 / 72.77 | 58.32 / 64.02 | 0.786 | 0.322 |
| ALMT | 23.71 | 24.97 | 61.84 / 59.67 | 65.30 / **65.19** | 1.266 | 0.336 | 34.78 | 34.95 | **71.62** / 73.98 | 72.24 / 74.54 | **0.743** | 0.315 |
| LNLN | **27.79** | **31.19** | **65.95 / 65.01** | **65.95** / 65.14 | **1.219** | **0.383** | 42.22 | 42.56 | 71.55 / **74.74** | **73.49 / 77.40** | 0.762 | **0.408** |
| Random Missing Rate $r = 0.8$ | | | | | | | | | | | | |
| MISA | 19.92 | 20.99 | **63.56 / 62.24** | 63.16 / 61.67 | 1.311 | 0.321 | 33.29 | 29.51 | 63.43 / 71.30 | 49.95 / 59.69 | 0.927 | 0.267 |
| Self-MM | 19.29 | 22.11 | 59.55 / 58.26 | 49.98 / 53.56 | 1.313 | 0.282 | 38.69 | 38.78 | 65.07 / 71.83 | 54.44 / 61.49 | 0.805 | 0.259 |
| MMIM | 20.26 | 21.77 | 58.33 / 55.30 | 52.46 / 47.89 | 1.312 | 0.287 | 33.64 | 33.71 | 64.97 / 71.57 | 54.76 / 61.45 | 0.858 | 0.269 |
| CENET | 21.14 | 21.67 | 60.93 / 57.53 | 54.68 / 50.80 | 1.314 | 0.274 | **42.71** | **42.74** | 65.88 / 72.16 | 56.80 / 64.67 | 0.798 | 0.316 |
| TETFN | 22.01 | 23.76 | 59.40 / 56.85 | 48.73 / 45.59 | 1.337 | 0.274 | 22.01 | 42.11 | 59.40 / 56.85 | 48.73 / 45.59 | 1.337 | 0.274 |
| TFR-Net | 23.23 | 27.70 | 58.49 / 55.98 | 44.70 / 41.88 | 1.497 | 0.155 | 42.23 | 27.70 | 65.05 / 71.95 | 54.91 / 61.82 | 0.807 | 0.241 |
| ALMT | 23.13 | 23.66 | 60.37 / 58.31 | **65.45 / 66.14** | 1.310 | 0.273 | 34.01 | 34.09 | 68.15 / 71.48 | 69.12 / 72.28 | **0.774** | 0.231 |
| LNLN | **26.34** | **28.23** | 62.75 / 62.10 | 62.56 / 62.03 | **1.283** | **0.314** | 40.76 | 40.97 | 68.62 / 72.86 | 71.83 / 76.80 | 0.791 | **0.325** |
| Random Missing Rate $r = 0.9$ | | | | | | | | | | | | |
| MISA | 17.78 | 18.41 | 58.64 / **58.21** | 56.84 / 56.19 | 1.369 | **0.226** | 32.29 | 28.03 | 62.95 / 71.07 | 48.80 / 59.12 | 0.941 | 0.180 |
| Self-MM | 18.32 | 19.78 | 58.59 / 55.25 | 46.16 / 47.46 | 1.353 | 0.197 | 37.46 | 37.50 | 63.85 / 71.24 | 51.32 / 59.72 | 0.821 | 0.188 |
| MMIM | 18.95 | 19.53 | 55.29 / 51.65 | 47.33 / 40.89 | 1.357 | 0.186 | 32.61 | 32.67 | 63.69 / 71.10 | 51.26 / 59.99 | 0.877 | 0.197 |
| CENET | 19.15 | 19.10 | **58.99** / 54.76 | 50.01 / 46.58 | 1.357 | 0.181 | **42.08** | **42.08** | 64.14 / 70.42 | 54.27 / 62.33 | 0.814 | **0.254** |
| TETFN | 20.75 | 21.19 | 58.43 / 55.88 | 45.24 / 42.12 | 1.378 | 0.186 | 20.75 | 41.55 | 58.43 / 55.88 | 45.24 / 42.12 | 1.378 | 0.186 |
| TFR-Net | 21.67 | **25.12** | 57.93 / 55.44 | 43.01 / 40.18 | 1.534 | 0.155 | 41.73 | 25.12 | 64.91 / 71.34 | 52.02 / 59.99 | 0.820 | 0.175 |
| ALMT | 20.31 | 20.50 | 57.32 / 56.66 | **64.92 / 67.82** | 1.349 | 0.205 | 34.40 | 34.40 | 61.41 / 68.65 | 63.32 / 69.83 | 0.810 | 0.138 |
| LNLN | **22.98** | 23.86 | 56.50 / 56.51 | 56.32 / 56.47 | **1.349** | 0.202 | 40.10 | 40.19 | **64.83 / 71.51** | **70.60 / 77.52** | 0.820 | 0.221 |

Table 10: Details of robust comparison on SIMS with different random missing rates. Note: The smaller MAE indicates the better performance.

| Method | Acc-5 | Acc-3 | Acc-2 | F1 | MAE | Corr | Method | Acc-5 | Acc-3 | Acc-2 | F1 | MAE | Corr |
|---|---|---|---|---|---|---|---|---|---|---|---|---|---|
| Random Missing Rate $r = 0$ | | | | | | | Random Missing Rate $r = 0.5$ | | | | | | |
| MISA | 40.55 | 63.38 | 78.19 | 77.22 | 0.449 | 0.576 | MISA | 30.56 | 54.78 | 71.26 | 64.16 | 0.552 | 0.367 |
| Self-MM | 40.77 | 64.92 | 78.26 | 78.00 | **0.421** | 0.584 | Self-MM | 32.02 | 53.90 | 71.41 | 67.11 | 0.517 | 0.390 |
| MMIM | 37.42 | 60.69 | 75.42 | 73.10 | 0.475 | 0.528 | MMIM | 33.41 | 52.37 | 68.49 | 64.81 | 0.553 | 0.336 |
| CENET | 23.85 | 54.05 | 68.71 | 57.82 | 0.578 | 0.137 | CENET | 23.12 | 54.05 | 68.71 | 57.92 | 0.588 | 0.107 |
| TETFN | **41.94** | **65.86** | **80.23** | 79.25 | 0.424 | **0.589** | TETFN | 33.48 | 56.24 | 72.43 | 67.30 | **0.512** | 0.394 |
| TFR-Net | 33.85 | 54.12 | 69.15 | 58.44 | 0.562 | 0.254 | TFR-Net | 24.65 | 52.37 | 67.47 | 58.66 | 0.685 | 0.171 |
| ALMT | 23.41 | 54.78 | 75.64 | 76.27 | 0.527 | 0.536 | ALMT | 18.38 | 47.12 | 68.27 | 71.22 | 0.563 | 0.395 |
| LNLN | 38.66 | 63.97 | 75.93 | **79.89** | 0.458 | 0.570 | LNLN | **36.40** | **57.70** | **72.72** | **78.77** | 0.513 | **0.412** |
| Random Missing Rate $r = 0.1$ | | | | | | | Random Missing Rate $r = 0.6$ | | | | | | |
| MISA | 38.88 | 63.02 | 77.39 | 75.82 | 0.461 | 0.561 | MISA | 27.72 | 53.97 | 70.46 | 61.81 | 0.578 | 0.286 |
| Self-MM | 40.26 | 63.53 | 77.32 | 76.76 | 0.433 | 0.563 | Self-MM | 29.10 | 51.86 | 70.02 | 64.21 | 0.548 | 0.313 |
| MMIM | 37.27 | 60.90 | 74.25 | 72.08 | 0.473 | 0.529 | MMIM | 29.18 | 49.31 | 67.91 | 63.86 | 0.578 | 0.270 |
| CENET | 22.83 | 53.98 | 68.57 | 57.36 | 0.580 | 0.136 | CENET | 22.46 | 53.69 | 69.00 | 58.64 | 0.592 | 0.102 |
| TETFN | **41.36** | **64.62** | **78.92** | 77.70 | **0.432** | **0.578** | TETFN | 29.90 | 53.54 | 70.97 | 64.19 | 0.545 | 0.309 |
| TFR-Net | 30.12 | 53.25 | 68.85 | 59.38 | 0.596 | 0.203 | TFR-Net | 24.80 | 52.59 | 67.03 | 58.30 | 0.696 | 0.157 |
| ALMT | 22.10 | 55.14 | 74.40 | 75.19 | 0.530 | 0.537 | ALMT | 18.67 | 43.69 | 66.81 | 70.69 | 0.574 | 0.322 |
| LNLN | 38.51 | 62.73 | 76.29 | **80.07** | 0.458 | 0.562 | LNLN | **33.70** | **54.63** | **71.55** | **79.56** | **0.535** | **0.352** |
| Random Missing Rate $r = 0.2$ | | | | | | | Random Missing Rate $r = 0.7$ | | | | | | |
| MISA | 38.15 | 59.23 | 74.33 | 71.70 | 0.489 | 0.490 | MISA | 24.87 | 52.52 | **69.95** | 59.54 | 0.601 | 0.167 |
| Self-MM | 38.37 | 61.71 | 74.98 | 73.71 | 0.464 | 0.500 | Self-MM | 25.53 | 50.62 | 69.58 | 62.28 | 0.571 | 0.198 |
| MMIM | 37.27 | 57.33 | 72.36 | 69.80 | 0.504 | 0.460 | MMIM | 28.59 | 46.53 | 66.89 | 62.23 | 0.595 | 0.190 |
| CENET | 22.25 | 54.20 | 68.57 | 57.64 | 0.583 | 0.132 | CENET | 21.81 | **53.32** | 67.69 | 57.87 | 0.599 | 0.070 |
| TETFN | **39.46** | 61.56 | **75.49** | 73.59 | **0.457** | **0.527** | TETFN | 27.86 | 51.06 | 69.29 | 61.09 | 0.572 | 0.190 |
| TFR-Net | 29.03 | 53.61 | 68.64 | 59.74 | 0.619 | 0.191 | TFR-Net | 23.78 | 52.30 | 67.18 | 58.15 | 0.707 | 0.163 |
| ALMT | 21.08 | 53.17 | 72.65 | 73.90 | 0.541 | 0.485 | ALMT | 18.02 | 38.66 | 65.57 | 70.27 | 0.586 | 0.218 |
| LNLN | 38.88 | **61.78** | 74.76 | **78.53** | 0.474 | 0.513 | LNLN | **30.27** | 51.64 | 69.73 | **79.37** | **0.558** | **0.261** |
| Random Missing Rate $r = 0.3$ | | | | | | | Random Missing Rate $r = 0.8$ | | | | | | |
| MISA | 36.40 | 59.30 | 74.11 | 70.40 | 0.505 | 0.464 | MISA | 22.69 | 52.22 | 69.37 | 57.82 | 0.610 | 0.092 |
| Self-MM | 37.93 | 59.81 | 74.76 | 72.85 | 0.474 | 0.487 | Self-MM | 22.03 | 50.77 | 69.51 | 60.68 | 0.585 | 0.138 |
| MMIM | 37.71 | 58.06 | 72.36 | 69.52 | 0.512 | 0.436 | MMIM | 22.32 | 44.35 | 65.28 | 60.53 | 0.607 | 0.145 |
| CENET | 21.44 | 54.05 | 68.42 | 57.41 | 0.578 | 0.175 | CENET | 21.73 | 52.15 | 67.47 | 58.44 | 0.599 | 0.074 |
| TETFN | **38.80** | **61.92** | **75.86** | 73.28 | **0.463** | **0.521** | TETFN | 23.48 | 49.60 | **69.88** | 60.92 | 0.584 | 0.154 |
| TFR-Net | 27.64 | 52.30 | 68.42 | 59.88 | 0.640 | 0.182 | TFR-Net | 22.97 | **52.74** | 67.54 | 57.55 | 0.721 | 0.100 |
| ALMT | 20.35 | 50.62 | 72.06 | 73.64 | 0.546 | 0.469 | ALMT | 18.60 | 34.06 | 64.19 | 69.64 | 0.597 | 0.133 |
| LNLN | 38.37 | 60.98 | 74.25 | **78.60** | 0.478 | 0.509 | LNLN | **27.94** | 50.47 | 69.58 | **80.23** | **0.580** | **0.183** |
| Random Missing Rate $r = 0.4$ | | | | | | | Random Missing Rate $r = 0.9$ | | | | | | |
| MISA | 34.86 | 57.33 | 72.87 | 67.52 | 0.523 | 0.436 | MISA | 20.64 | 52.95 | **69.22** | 57.01 | 0.617 | 0.041 |
| Self-MM | 34.57 | 58.28 | 73.30 | 70.36 | 0.482 | 0.479 | Self-MM | 22.17 | 52.15 | 68.92 | 58.32 | **0.586** | 0.111 |
| MMIM | 34.57 | 55.36 | 69.95 | 66.49 | 0.533 | 0.399 | MMIM | 20.35 | 42.67 | 65.72 | 59.64 | 0.610 | 0.096 |
| CENET | 22.54 | 54.12 | 68.49 | 57.68 | 0.583 | 0.141 | CENET | 20.86 | 48.07 | 65.72 | 58.18 | 0.609 | -0.002 |
| TETFN | 35.81 | 58.93 | **73.81** | 70.66 | **0.473** | **0.504** | TETFN | 22.10 | 45.73 | 68.92 | 58.75 | 0.590 | 0.108 |
| TFR-Net | 25.31 | 51.86 | 67.91 | 59.16 | 0.664 | 0.176 | TFR-Net | 23.05 | **53.76** | 69.08 | 57.71 | 0.721 | 0.088 |
| ALMT | 19.91 | 49.45 | 70.75 | 72.97 | 0.549 | 0.470 | ALMT | 19.47 | 26.91 | 66.23 | 73.76 | 0.596 | 0.076 |
| LNLN | **37.49** | **60.03** | **73.81** | **78.82** | 0.491 | 0.481 | LNLN | **26.19** | 47.48 | 68.64 | **80.42** | 0.591 | **0.127** |

Self-MM predicts all samples as weak negative when the missing rate is 0.9. Although the accuracy of Self-MM is high when the missing rate $r$ is high, it does not demonstrate that the model is more robust.

In addition, TETFN and TFR-Net tend to randomly predict within the negative and weak negative categories under high noise, indicating that their lazy behavior is not as severe as methods like Self-MM, MMIM, and MISA. Unlike other methods, although LNLN also suffers from the impact of data missing, the model's lazy behavior is less severe. For example, at a missing rate of 0.9, the model is still attempting to predict the samples, and there is no case of predictions favoring a particular category to an extreme extent.

From Figure 6 and Figure 7, we can see that a similar phenomenon also occurs in the MOSEI and SIMS datasets. These observations demonstrate that methods targeting complete data often fail when modeling highly incomplete data. Therefore, the design of methods specifically for modeling incomplete data is necessary.

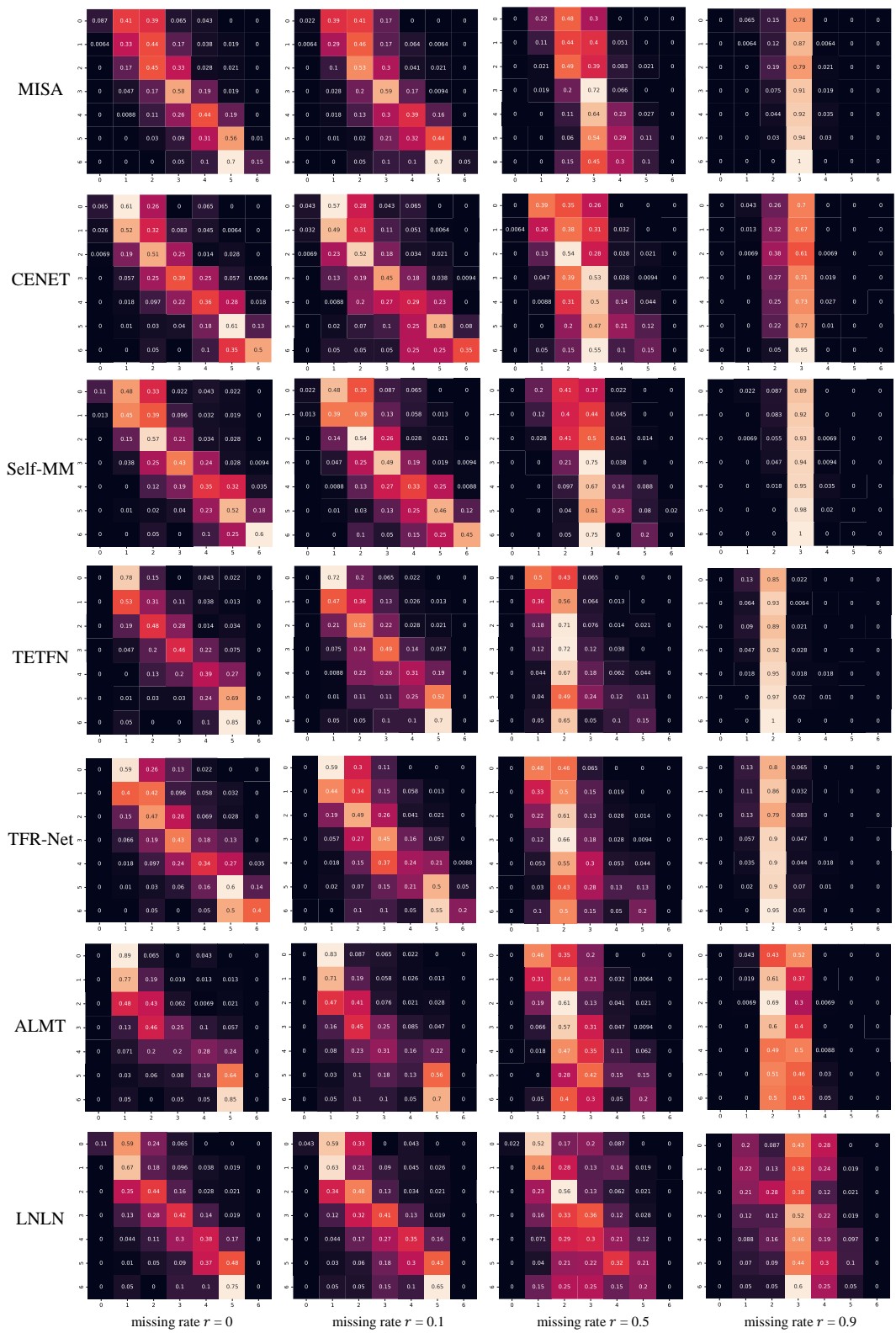

Figure 5: Seven-category confusion matrix of several representative methods on MOSI dataset. Note: 0-6 denote strongly negative, weakly negative, negative, neutral, weakly positive, positive, and strongly positive, respectively.

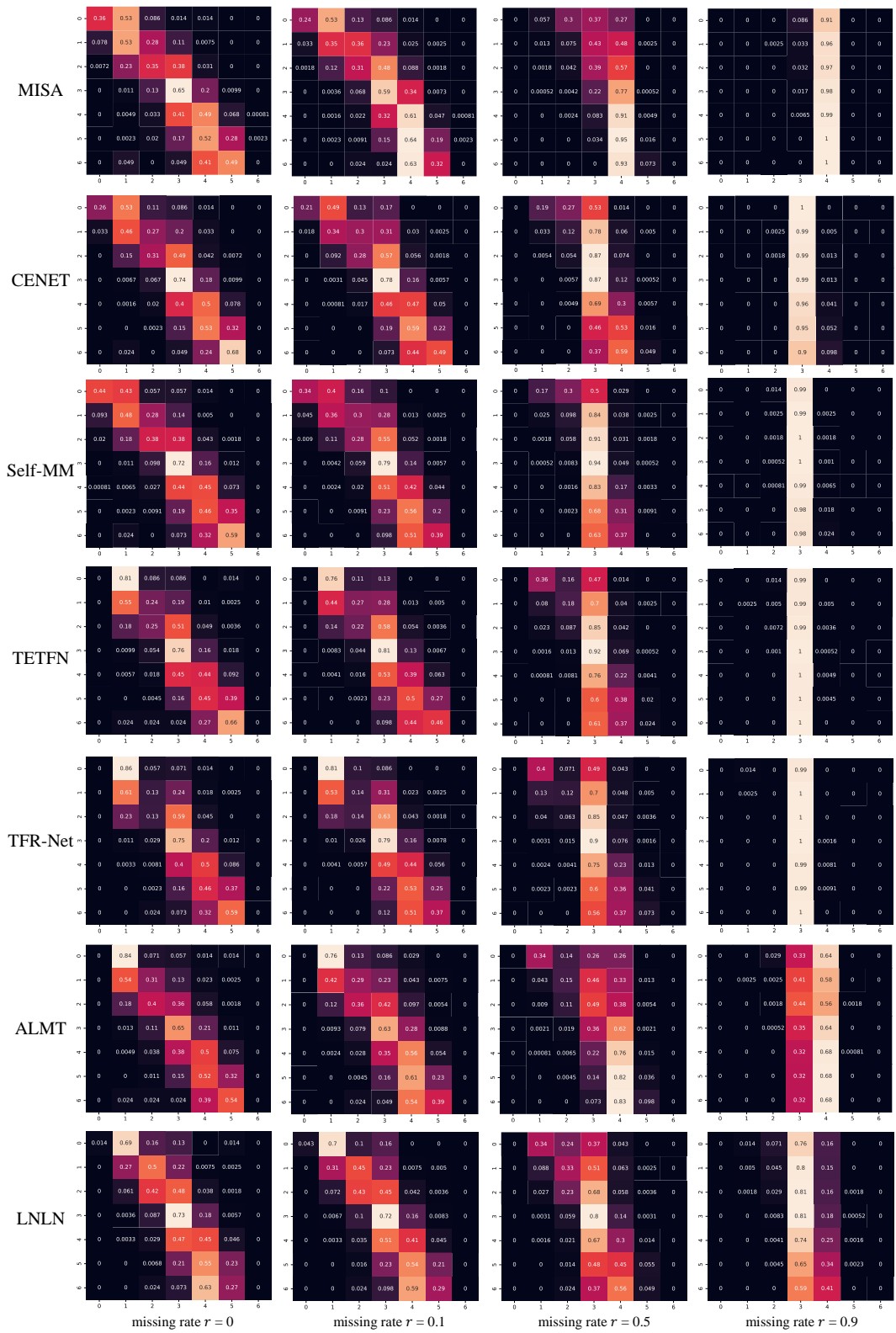

Figure 6: Seven-category confusion matrix of several representative methods on MOSEI dataset. Note: 0-6 denote strongly negative, weakly negative, negative, neutral, weakly positive, positive, and strongly positive, respectively.

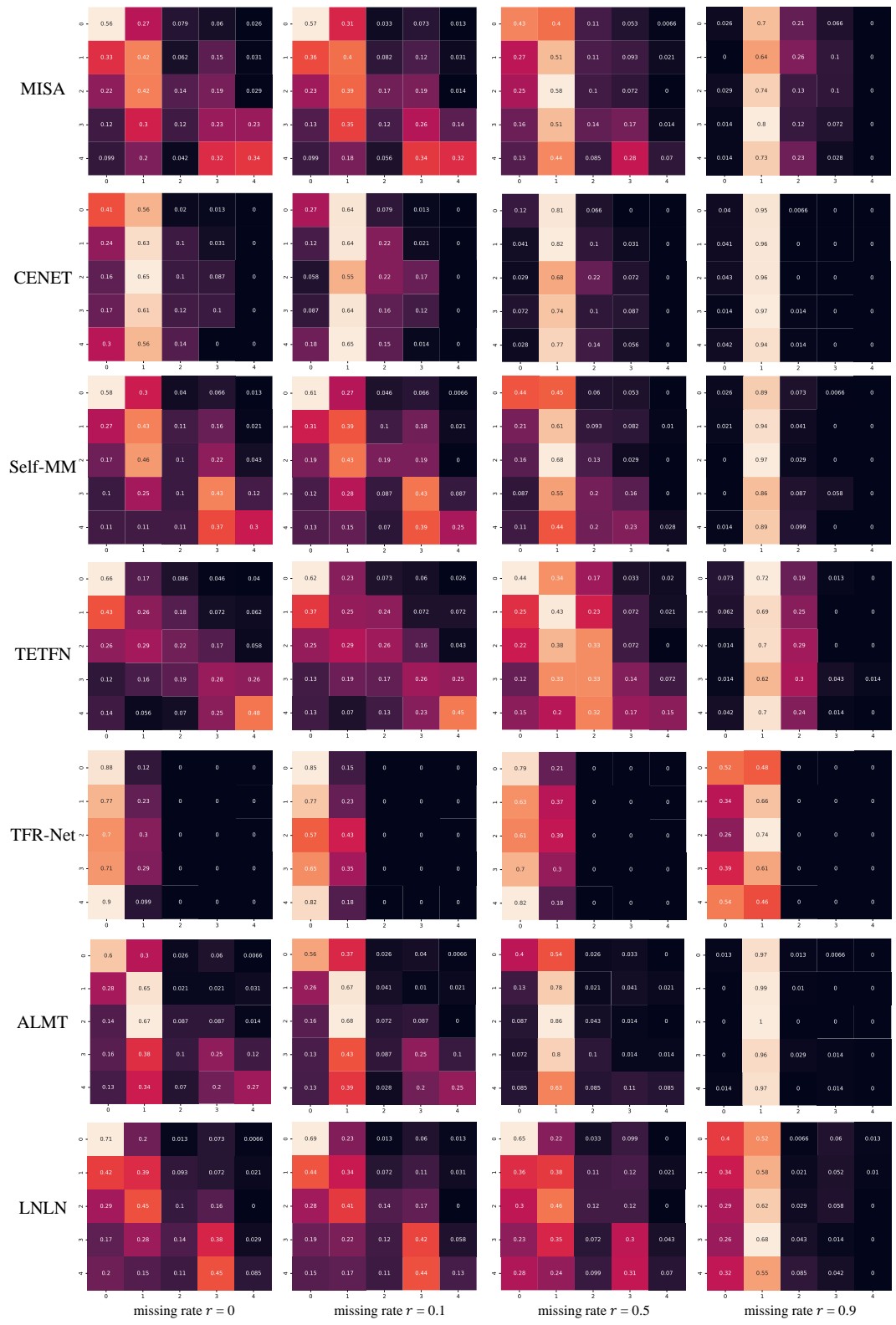

Figure 7: Five-category confusion matrix of several representative methods on SIMS dataset. Note: 0-4 denote weakly negative, negative, neutral, weakly positive, and positive, respectively.

## A.7 Visualization of Representations

Figure 8 shows the language feature $H_l^1$ extracted without applying data missing and the corrected dominant feature $H_d^1$ with different data missing rates. we can see that the model tends to learn $H_d^1$, which is consistent with the distribution of the $H_l^1$, indicating that LNLN can complement the information of the language modality when data is missing. Additionally, when the missing rate $r$ is 1.0 (no valid information for the input), it becomes difficult for the model to learn $H_d^1$ that is consistent with the language feature. This also suggests that the generator is indeed trying to learn $H_d^1$ for fusion in DMML, rather than just guessing.

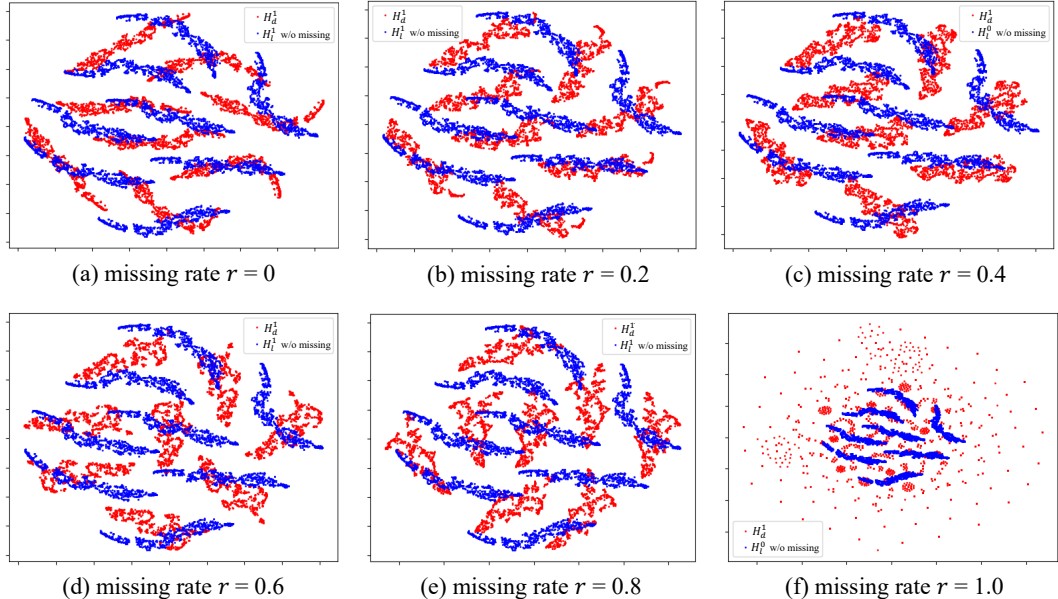

(a) missing rate $r = 0$      (b) missing rate $r = 0.2$      (c) missing rate $r = 0.4$

(d) missing rate $r = 0.6$      (e) missing rate $r = 0.8$      (f) missing rate $r = 1.0$

Figure 8: Visualization of the corrected dominant feature $H_d^1$ with different data missing rate and the language feature $H_l^1$ without data missing.

## A.8 Analysis of the generalization to modality missing scenarios

To explore the generalization of the method, we further evaluated it in the special case of data missing, *i.e.,* modality missing scenarios. The overall performance of the model is shown in Table 11 and Table 12. The detailed performance of the model is shown in Table 13 and Table 14. It is evident that LNLN, ALMT, TETFN, and Self-MM all exhibit excellent generalization.

We believe that the Dominant Modality Correction (DMC) proposed in LNLN, the Adaptive Hyper-modality Learning (AHL) proposed in ALMT, and the Unimodal Label Generation Module (ULGM) proposed/used in Self-MM and TETFN deserve more research attention in the future. These modules and their corresponding ideas have significant reference value for achieving robust Multimodal Sentiment Analysis (MSA).

However, we also found that many methods converge to the same value when the noise is high. This is similar to the phenomenon observed in previous data missing scenarios, where many models exhibited lazy behavior. Therefore, handling modality missing and improving model robustness in different noise scenarios remain critical challenges that need to be addressed in the field of MSA.

Table 11: Generalization comparison of the overall performance on MOSI and MOSEI with random modality missing. Note: the parameters used for evaluation are consistent with those used for testing in random data missing. The smaller MAE indicates better performance.

| Method | MOSI | | | | | | MOSEI | | | | | |
|---|---|---|---|---|---|---|---|---|---|---|---|---|
| | Acc-7 | Acc-5 | Acc-2 | F1 | MAE | Corr | Acc-7 | Acc-5 | Acc-2 | F1 | MAE | Corr |
| MISA | 29.44 | 31.76 | 67.64 / 67.27 | 64.81 / 64.30 | 1.096 | **0.461** | 41.45 | 40.32 | 73.94 / 77.49 | 66.65 / 71.30 | 0.752 | 0.438 |
| Self-MM | 29.74 | **36.40** | **71.45 / 68.17** | 63.71 / 61.91 | **1.052** | 0.450 | 45.17 | 46.05 | 74.12 / **77.85** | 67.00 / 71.81 | **0.684** | 0.447 |
| MMIM | 31.28 | 33.74 | 68.34 / 65.12 | 63.38 / 58.28 | 1.058 | 0.444 | 41.36 | 42.53 | 73.19 / 76.14 | 68.11 / 70.18 | 0.732 | 0.448 |
| CENET | 30.30 | 33.94 | 70.37 / 66.67 | 62.66 / 58.61 | 1.068 | 0.450 | 47.41 | 48.19 | **73.45** / 74.14 | 68.43 / 70.37 | **0.684** | **0.480** |
| TETFN | **32.66** | 36.26 | 70.17 / 68.15 | 62.47 / 60.18 | 1.061 | 0.446 | **47.52** | **48.41** | 73.87 / 77.43 | 66.70 / 71.45 | 0.691 | 0.426 |
| TFR-Net | 29.76 | 34.96 | 70.49 / 67.95 | 62.87 / 60.08 | 1.177 | 0.451 | 47.33 | 48.08 | 73.50 / 77.45 | 66.89 / 71.28 | 0.700 | 0.427 |
| ALMT | 27.87 | 30.73 | 70.53 / 67.47 | **75.89 / 73.35** | 1.130 | 0.447 | 25.94 | 27.46 | 60.66 / 63.37 | 70.63 / 71.24 | 0.711 | 0.354 |
| LNLN | 31.93 | 34.86 | 68.53 / 65.64 | 72.01 / 69.65 | 1.093 | 0.423 | 43.85 | 44.50 | 73.52 / 77.04 | **80.85 / 83.19** | 0.733 | 0.425 |

Table 12: Generalization comparison of the overall performance on SIMS dataset with random modality missing. Note: the parameters used for evaluation are consistent with those used for testing in random data missing. The smaller MAE indicate the better performance.

| Method | Acc-5 | Acc-3 | Acc-2 | F1 | MAE | Corr |
|---|---|---|---|---|---|---|
| MISA | 29.62 | 56.82 | 73.61 | 66.32 | 0.535 | **0.330** |
| Self-MM | 30.36 | **59.56** | 75.01 | 73.13 | **0.508** | 0.321 |
| MMIM | 27.21 | 50.79 | 71.52 | 65.48 | 0.546 | 0.286 |
| CENET | 19.95 | 43.63 | 64.08 | 56.96 | 0.601 | 0.033 |
| TETFN | **30.61** | 50.98 | **74.77** | 68.00 | 0.514 | 0.317 |
| TFR-Net | 24.69 | 52.57 | 69.31 | 58.03 | 0.629 | 0.133 |
| ALMT | 21.82 | 39.39 | 72.30 | 78.86 | 0.563 | 0.287 |
| LNLN | 30.39 | 54.32 | 72.67 | **80.87** | 0.527 | 0.287 |

Table 13: Generalization comparison on MOSI and MOSEI with random modality missing. Note: The smaller MAE indicates the better performance.

| Method | MOSI | | | | | | MOSEI | | | | | |
|---|---|---|---|---|---|---|---|---|---|---|---|---|
| | Acc-7 | Acc-5 | Acc-2 | F1 | MAE | Corr | Acc-7 | Acc-5 | Acc-2 | F1 | MAE | Corr |
| Language Modality Missing | | | | | | | | | | | | |
| MISA | 15.60 | 15.65 | 52.18 / 53.74 | 48.09 / 49.42 | 1.422 | **0.171** | 30.93 | 26.75 | 62.85 / **71.02** | 48.51 / 58.99 | 0.946 | 0.128 |
| Self-MM | 16.86 | 19.29 | 57.77 / 54.37 | 42.31 / 43.81 | 1.381 | 0.129 | 36.59 | 36.59 | 63.03 / **71.02** | 49.22 / 58.99 | 0.832 | 0.152 |
| MMIM | 17.44 | 17.20 | 54.67 / 49.61 | 46.95 / 37.63 | **1.363** | 0.153 | 31.96 | 31.97 | 62.71 / 71.01 | 54.70 / 59.43 | 0.886 | 0.195 |
| CENET | 17.54 | 17.54 | 57.67 / 51.80 | 42.26 / 35.71 | 1.387 | 0.118 | **41.70** | **41.70** | **63.23** / 69.38 | 52.41 / 60.67 | **0.821** | **0.238** |
| TETFN | 21.19 | 21.14 | 57.77 / 55.25 | 42.31 / 39.32 | 1.403 | 0.097 | 41.31 | 41.31 | 62.82 / 71.01 | 48.53 / 58.98 | 0.831 | 0.148 |
| TFR-Net | **23.13** | **25.95** | **57.82** / **55.30** | 42.43 / 39.43 | 1.563 | 0.167 | 41.40 | 41.43 | 62.85 / **71.02** | 48.88 / 58.99 | 0.829 | 0.163 |
| ALMT | 19.29 | 19.68 | 56.40 / 55.05 | **66.99** / **66.01** | 1.394 | 0.139 | 21.58 | 21.58 | 54.28 / 57.38 | 69.52 / 70.01 | 0.867 | 0.071 |
| LNLN | 18.80 | 17.68 | 52.18 / 49.03 | 58.89 / 56.84 | 1.427 | 0.075 | 39.10 | 39.10 | 62.85 / **71.02** | **77.19** / **83.06** | 0.847 | 0.156 |
| Audio Modality Missing | | | | | | | | | | | | |
| MISA | 43.49 | 48.44 | 83.74 / 81.58 | 83.72 / 81.49 | 0.762 | 0.776 | 51.59 | 53.91 | 84.94 / 83.91 | 84.62 / 83.46 | 0.561 | 0.759 |
| Self-MM | 42.81 | **52.33** | **85.22** / **83.24** | **85.19** / **83.26** | 0.720 | 0.790 | 53.91 | 55.66 | 85.31 / **84.70** | 85.06 / **84.64** | 0.532 | 0.763 |
| MMIM | **45.39** | 49.52 | 83.08 / 81.15 | 83.00 / 81.02 | 0.744 | 0.777 | 51.01 | 53.26 | 83.72 / 81.51 | 83.60 / 81.29 | 0.572 | 0.726 |
| CENET | 43.05 | 50.39 | 83.08 / 81.54 | 83.06 / 81.54 | 0.750 | 0.785 | **54.34** | 55.98 | **85.37** / 82.08 | **85.35** / 82.44 | **0.531** | **0.770** |
| TETFN | 44.07 | 51.31 | 82.52 / 81.00 | 82.57 / 80.99 | **0.719** | **0.794** | 54.15 | 55.94 | 85.08 / 83.99 | 85.05 / 84.05 | 0.543 | 0.753 |
| TFR-Net | 37.71 | 45.48 | 82.72 / 80.52 | 82.71 / 80.45 | 0.821 | 0.759 | 53.75 | 55.34 | 84.95 / 84.65 | 84.69 / 84.32 | 0.550 | 0.745 |
| ALMT | 36.49 | 41.84 | 84.60 / 79.88 | 84.81 / 80.70 | 0.865 | 0.767 | 30.51 | 33.55 | 67.13 / 70.05 | 71.76 / 72.59 | 0.535 | 0.556 |
| LNLN | 45.05 | 52.04 | 84.86 / 82.21 | 85.12 / 82.43 | 0.760 | 0.772 | 50.63 | 51.92 | 84.39 / 83.09 | 84.71 / 83.34 | 0.610 | 0.736 |
| Visual Modality Missing | | | | | | | | | | | | |
| MISA | 43.29 | 47.09 | 82.37 / 81.00 | 82.43 / 80.99 | 0.777 | 0.777 | 51.76 | 53.80 | **85.25** / 83.99 | **85.11** / 83.76 | 0.550 | 0.757 |
| Self-MM | 42.76 | **55.47** | **85.11** / **82.80** | 85.09 / **82.82** | 0.722 | 0.789 | **53.68** | **55.47** | 85.20 / **84.62** | 84.95 / **84.59** | **0.533** | 0.761 |
| MMIM | 44.75 | 49.85 | 83.54 / 81.83 | 83.52 / 81.78 | 0.740 | 0.778 | 50.45 | 52.86 | 83.71 / 81.31 | 83.53 / 81.09 | 0.579 | 0.722 |
| CENET | 43.20 | 50.39 | 83.08 / 81.49 | 83.06 / 81.48 | 0.748 | 0.785 | 52.95 | 54.48 | 84.99 / 81.04 | 85.03 / 81.56 | 0.544 | **0.761** |
| TETFN | 44.22 | 51.46 | 82.62 / 81.10 | 82.67 / 81.09 | **0.719** | **0.794** | 53.55 | 55.34 | 84.84 / 83.78 | 84.81 / 83.86 | 0.544 | **0.761** |
| TFR-Net | 37.85 | 45.87 | 83.54 / 81.10 | 83.41 / 80.95 | 0.799 | 0.760 | 52.94 | 54.45 | 83.97 / 83.48 | 83.93 / 83.18 | 0.577 | 0.726 |
| ALMT | 36.39 | 41.93 | 84.81 / 79.98 | 84.81 / 80.76 | 0.863 | 0.768 | 29.80 | 32.95 | 66.81 / 69.41 | 71.72 / 72.23 | 0.559 | 0.747 |
| LNLN | **45.05** | 51.99 | 84.91 / 82.26 | **85.17** / 82.48 | 0.759 | 0.772 | 50.16 | 51.44 | 84.09 / 83.41 | 84.43 / 83.76 | 0.577 | 0.729 |
| Language & Audio Modality Missing | | | | | | | | | | | | |
| MISA | 15.55 | 15.60 | 55.29 / 55.20 | 49.53 / 49.24 | 1.419 | 0.098 | 31.17 | 26.76 | 62.85 / **71.02** | 48.51 / 58.99 | 0.961 | 0.121 |
| Self-MM | 16.86 | 19.29 | 57.77 / 54.03 | 42.31 / 43.81 | **1.381** | **0.130** | 36.61 | 36.61 | **63.01** / **71.02** | 49.18 / 58.99 | 0.831 | 0.144 |
| MMIM | 17.59 | 17.40 | 52.24 / 48.20 | 39.57 / 31.53 | 1.389 | 0.024 | 31.91 | 31.92 | 62.31 / 70.94 | 54.32 / 59.28 | 0.886 | 0.182 |
| CENET | 17.40 | 17.40 | 57.67 / 51.85 | 42.26 / 35.65 | 1.387 | 0.110 | **41.40** | **41.40** | 60.69 / 64.21 | 51.22 / 56.68 | **0.823** | **0.236** |
| TETFN | 21.19 | 21.14 | 57.77 / 55.25 | 42.31 / 39.32 | 1.403 | 0.098 | 41.31 | 41.31 | 62.82 / 71.00 | 48.51 / 58.98 | 0.832 | 0.144 |
| TFR-Net | **21.38** | **23.96** | **58.03** / **55.59** | 43.37 / 40.54 | 1.554 | 0.107 | 41.37 | 41.38 | 62.81 / **71.02** | 48.75 / 58.99 | 0.829 | 0.161 |
| ALMT | 19.24 | 19.24 | 56.35 / 54.96 | **66.89** / **65.86** | 1.397 | 0.104 | 21.57 | 21.57 | 54.28 / 57.28 | 69.52 / 70.09 | 0.870 | 0.091 |
| LNLN | 18.80 | 17.68 | 52.18 / 49.03 | 58.89 / 56.84 | 1.427 | 0.072 | 34.72 | 34.72 | 62.85 / **71.02** | **77.19** / **83.06** | 0.900 | 0.145 |
| Language & Visual Modality Missing | | | | | | | | | | | | |
| MISA | 15.45 | 15.40 | 48.98 / 50.87 | 41.79 / 43.51 | 1.427 | **0.169** | 31.57 | 26.89 | 62.85 / **71.02** | 48.51 / 58.99 | 0.936 | 0.105 |
| Self-MM | 16.47 | 19.24 | **57.77** / 51.75 | 42.31 / 35.44 | 1.386 | 0.072 | 36.55 | 36.55 | 62.85 / **71.02** | 48.51 / 58.99 | **0.838** | 0.101 |
| MMIM | 17.93 | 17.40 | 53.05 / 48.64 | 43.80 / 36.52 | **1.366** | 0.155 | 32.12 | 32.12 | **62.88** / 70.94 | 48.91 / 59.00 | 0.891 | **0.143** |
| CENET | 17.54 | 17.54 | 57.67 / 51.80 | 42.26 / 35.71 | 1.387 | 0.118 | **41.36** | **41.36** | 61.78 / 67.49 | 51.87 / 59.63 | **0.838** | 0.112 |
| TETFN | 21.19 | 21.14 | **57.77** / **55.25** | 42.31 / 39.32 | 1.403 | 0.097 | **41.36** | **41.36** | 62.81 / **71.02** | 48.51 / 58.99 | 0.840 | 0.017 |
| TFR-Net | **22.50** | **25.07** | **57.77** / **55.25** | 42.31 / 39.32 | 1.486 | 0.154 | **41.36** | **41.36** | 62.47 / **71.02** | 51.16 / 58.99 | 0.839 | 0.039 |
| ALMT | 19.29 | 19.77 | 56.45 / 55.10 | **67.09** / **66.11** | 1.394 | 0.136 | 21.54 | 21.54 | 54.28 / 57.01 | 69.52 / 70.35 | 0.874 | -0.086 |
| LNLN | 18.80 | 17.68 | 52.18 / 49.03 | 58.89 / 56.84 | 1.427 | 0.075 | 38.41 | 38.41 | 62.85 / **71.02** | **77.19** / **83.06** | 0.853 | 0.052 |
| Audio & Visual Modality Missing | | | | | | | | | | | | |
| MISA | 43.25 | 48.40 | 83.28 / 81.24 | 83.29 / 81.17 | 0.768 | 0.776 | 51.67 | 53.80 | 84.88 / 83.97 | 84.64 / 83.64 | 0.558 | 0.756 |
| Self-MM | 42.71 | **52.77** | **85.06** / **82.80** | 85.04 / **82.82** | 0.722 | 0.789 | **53.67** | **55.43** | **85.32** / **84.74** | **85.06** / **84.67** | **0.535** | **0.761** |
| MMIM | 44.56 | 51.07 | 83.48 / 81.29 | 83.42 / 81.17 | 0.748 | 0.777 | 50.70 | 53.07 | 83.79 / 81.14 | 83.61 / 80.96 | 0.579 | 0.722 |
| CENET | 43.05 | 50.39 | 83.08 / 81.54 | 83.06 / 81.54 | 0.750 | 0.785 | 52.71 | 54.22 | 84.64 / 80.66 | 84.71 / 81.23 | 0.545 | 0.760 |
| TETFN | 44.12 | 51.36 | 82.57 / 81.05 | 82.62 / 81.04 | **0.719** | 0.794 | 53.46 | 55.22 | 84.83 / 83.79 | 84.80 / 83.85 | 0.549 | 0.747 |
| TFR-Net | 36.00 | 43.44 | 83.08 / 79.98 | 83.01 / 79.82 | 0.838 | 0.758 | 53.13 | 54.54 | 83.97 / 83.49 | 83.91 / 83.18 | 0.578 | 0.726 |
| ALMT | 36.49 | 41.93 | 84.55 / 79.83 | 84.76 / 80.65 | 0.864 | 0.767 | 30.64 | 33.57 | 67.14 / 69.11 | 71.77 / 72.15 | 0.560 | 0.748 |
| LNLN | **45.10** | 52.04 | 84.86 / 82.26 | **85.12** / 82.48 | 0.760 | 0.772 | 50.10 | 51.38 | 84.10 / 82.69 | 84.39 / 82.86 | 0.609 | 0.730 |

Table 14: Generalization comparison on SIMS with random modality missing. Note: The smaller MAE indicates the better performance.

| Method | Acc-5 | Acc-3 | Acc-2 | F1 | MAE | Corr | Method | Acc-5 | Acc-3 | Acc-2 | F1 | MAE | Corr |
|---|---|---|---|---|---|---|---|---|---|---|---|---|---|
| Language Modality Missing | | | | | | | Language & Audio Modalities Missing | | | | | | |
| MISA | 18.60 | 50.77 | 69.23 | 56.75 | 0.595 | **0.104** | MISA | 21.23 | **54.27** | 69.37 | 56.82 | 0.613 | **0.064** |
| Self-MM | 19.77 | 54.20 | 78.26 | 78.00 | **0.594** | 0.058 | Self-MM | 19.69 | 54.20 | **74.76** | 72.85 | **0.594** | 0.059 |
| MMIM | 17.22 | 41.28 | 67.76 | 57.76 | 0.617 | 0.055 | MMIM | 17.22 | 41.28 | 67.91 | 57.86 | 0.617 | 0.026 |
| CENET | 20.71 | 49.24 | 68.20 | 57.67 | 0.601 | 0.037 | CENET | 20.42 | 49.60 | 66.31 | 57.49 | 0.601 | 0.002 |
| TETFN | 19.26 | 36.11 | **69.37** | 56.82 | 0.603 | 0.045 | TETFN | 19.26 | 36.11 | 69.37 | 56.82 | 0.603 | 0.040 |
| TFR-Net | **22.25** | **54.27** | 69.30 | 56.78 | 0.712 | 0.008 | TFR-Net | 21.96 | 46.98 | 69.44 | 57.12 | 0.635 | 0.041 |
| ALMT | 21.15 | 24.95 | **69.37** | **81.91** | 0.595 | 0.049 | ALMT | 21.30 | 16.05 | 69.37 | **81.91** | **0.594** | 0.039 |
| LNLN | 21.37 | 45.30 | **69.37** | **81.91** | 0.597 | 0.005 | LNLN | **21.37** | 43.25 | 69.37 | **81.91** | 0.598 | -0.008 |
| Audio Modality Missing | | | | | | | Language & Visual Modalities Missing | | | | | | |
| MISA | 38.37 | 61.92 | 77.68 | 75.17 | 0.466 | 0.574 | MISA | 19.70 | 48.58 | 69.37 | 56.95 | 0.600 | **0.091** |
| Self-MM | 40.92 | 64.99 | 77.32 | 76.76 | **0.421** | 0.585 | Self-MM | 19.48 | **53.98** | **73.30** | 70.36 | **0.594** | 0.056 |
| MMIM | 37.27 | 60.61 | 75.20 | 72.96 | 0.475 | 0.527 | MMIM | 17.22 | 40.99 | 67.40 | 57.70 | 0.617 | 0.055 |
| CENET | 21.59 | 53.90 | 68.93 | 57.45 | 0.582 | 0.141 | CENET | 18.45 | 33.55 | 65.50 | 59.55 | 0.612 | -0.054 |
| TETFN | **41.94** | **65.86** | **80.16** | 79.16 | 0.424 | **0.589** | TETFN | 19.33 | 36.11 | 69.37 | 56.82 | 0.603 | 0.045 |
| TFR-Net | 26.40 | 52.45 | 69.66 | 59.21 | 0.569 | 0.239 | TFR-Net | 19.55 | 54.27 | 69.37 | 56.82 | 0.720 | 0.027 |
| ALMT | 23.05 | 59.45 | 75.27 | 76.36 | 0.531 | 0.529 | ALMT | 19.99 | 20.57 | 69.15 | 81.08 | 0.596 | 0.043 |
| LNLN | 39.53 | 64.19 | 75.78 | **79.69** | 0.454 | 0.570 | LNLN | **21.44** | 46.39 | 69.37 | **81.91** | 0.600 | 0.014 |
| Viusal Modality Missing | | | | | | | Audio & Visual Modalities Missing | | | | | | |
| MISA | 40.84 | 63.67 | 78.19 | 77.07 | 0.458 | 0.574 | MISA | 38.95 | 61.70 | 77.83 | 75.15 | 0.478 | 0.573 |
| Self-MM | 41.21 | 64.99 | 74.98 | 73.71 | **0.421** | 0.584 | Self-MM | 41.06 | 64.99 | 71.41 | 67.11 | **0.421** | 0.585 |
| MMIM | 37.34 | 60.47 | 75.57 | 73.43 | 0.475 | 0.528 | MMIM | 36.98 | 60.10 | 75.27 | 73.18 | 0.476 | 0.527 |
| CENET | 19.70 | 38.51 | 59.08 | 56.38 | 0.600 | 0.027 | CENET | 18.82 | 36.98 | 56.46 | 53.23 | 0.608 | 0.045 |
| TETFN | **41.94** | **65.86** | **80.16** | 79.19 | 0.424 | **0.589** | TETFN | **41.94** | **65.86** | **80.16** | 79.19 | 0.424 | **0.589** |
| TFR-Net | 30.20 | 54.56 | 69.15 | 58.86 | 0.566 | 0.247 | TFR-Net | 27.79 | 52.88 | 68.93 | 59.40 | 0.571 | 0.234 |
| ALMT | 22.83 | 55.51 | 75.35 | 75.75 | 0.529 | 0.535 | ALMT | 22.61 | 59.81 | 75.27 | 76.17 | 0.532 | 0.529 |
| LNLN | 38.80 | 63.31 | 76.22 | **80.11** | 0.458 | 0.569 | LNLN | 39.82 | 63.46 | 75.93 | **79.72** | 0.454 | 0.569 |

## B  Social Impacts

Our proposed LNLN has a wide range of applications in real-world scenarios, such as healthcare and human-computer interaction. However, it might also be misused to monitor individuals based on their affections for illegal purposes, potentially posing negative social impacts.

