# OpenReview forum: "Towards Robust Multimodal Sentiment Analysis with Incomplete Data"
_NeurIPS.cc/2024/Conference — NeurIPS 2024 poster_

### Official Review · Reviewer_LeNg · 2024-06-26

**Soundness:** 3
**Presentation:** 3
**Contribution:** 3
**Rating:** 7
**Confidence:** 3

**Summary:**

The paper proposes a Language-dominated Noise-resistant Learning Network (LNLN) to enhance the robustness of Multimodal Sentiment Analysis (MSA) under conditions of incomplete data. The model leverages the language modality, considered dominant due to its rich sentiment cues, to correct and reconstruct missing data from other modalities (visual and audio). Comprehensive experiments are conducted on datasets such as MOSI, MOSEI, and SIMS, demonstrating the superior performance of LNLN compared to existing baselines.

**Strengths:**

Comprehensive Evaluation: The experiments are thorough, covering multiple datasets and varying conditions of data incompleteness. This adds to the validity of the findings and allows for a fair comparison with existing methods.

Clear Motivation and Hypothesis: The paper clearly explains the rationale behind focusing on the language modality as the dominant one and provides a logical progression from problem identification to solution proposal.

Detailed Methodology: The methodological section is detailed, providing sufficient information for replication. The incorporation of adversarial learning and dynamic weighting strategies is well-explained.

Significant Performance Improvement: Empirical results indicate that LNLN consistently outperforms other methods, showcasing its robustness and effectiveness in real-world scenarios.

**Weaknesses:**

Real-World Application Scenarios: Discussing potential real-world applications and limitations of LNLN would enhance the paper's practical relevance.

**Questions:**

How to balance the hyperparameters of the loss function is not clear.

**Limitations:**

Discuss potential real-world applications and scenarios where LNLN can be particularly beneficial, as well as its limitations in these contexts.

---

> ### Author Rebuttal · Authors · 2024-08-05
>
> # Response to Reviewer LeNg
>
> ## **Response to W1**
>
> LNLN is well-suited for applications where multimodal data is often incomplete, which is a common challenge in many real-world s*cenarios*. For example, in platforms like Twitter, Instagram, and TikTok, users often express sentiments through a combination of text, images, videos, and audio. However, due to limitations such as incomplete captions, low-quality images, or background noise in videos, the data collected can be noisy or incomplete. LNLN's ability to prioritize the language modality while robustly integrating information from other modalities makes it ideal for analyzing such content, where data quality is variable. Moreover, in online learning platforms, students interact with content through text, video lectures, and audio discussions. LNLN can be used to analyze student sentiment and engagement, even when certain modalities, such as audio or video, are of poor quality or partially missing.
>
> However, limitations also exist. The data in real-world scenarios is much more complex. In addition to the presence of missing data, other factors need to be considered, such as diverse cultural contexts, varying user behavior patterns, and the influence of platform-specific features on the data. These factors can introduce additional noise and variability, which may require further model adaptation and tuning to handle effectively.
>
> In addition, from the experiments, it can be seen that the LNLN is able to better processing different levels of missing data noise. However, there are still some challenges and areas for improvement. **1)** There are differences in how well the model generalizes across different datasets, which suggests that further work is needed to ensure consistent performance. **2)** Tuning the hyperparameters, particularly those related to the loss functions, can be challenging and may require more sophisticated approaches to achieve optimal performance. **3)** Current MSA datasets are relatively small, and acquiring and annotating such data is difficult. Therefore, improving model performance with limited data is an important area of study. Given the strong performance of large language models in many domains, we believe that future research should explore MSA based on zero-shot and few-shot learning using large language models as a potential research direction.
>
> ## **Response to Q1**
>
> For the balance of hyperparameters, **we empirically selected them through experiments as described in Sections 3.6 and Sections 4.5.** Specifically, the hyperparameters of the loss function were tuned using a combination of grid search and validation performance. We first identified a range of reasonable values for each hyperparameter and then systematically evaluated their impact on a validation set, aiming to find the combination that optimally balances the trade-offs between different objectives.
>
> Additionally, **as shown in Table 1 of the uploaded PDF**, we have included detailed experimental results on the MOSI and SIMS datasets, further demonstrating the effectiveness of the loss function and the tuning process.

---

### Official Review · Reviewer_xa9F · 2024-06-28

**Soundness:** 3
**Presentation:** 3
**Contribution:** 3
**Rating:** 7
**Confidence:** 4

**Summary:**

The paper addresses data incompleteness in Multimodal Sentiment Analysis (MSA) by presenting the Language-dominated Noise-resistant Learning Network (LNLN). By considering language as the dominant modality, LNLN introduces a Dominant Modality Correction (DMC) module and Dominant Modality-Based Multimodal Learning (DMML) module to enhance robustness against noise. Extensive experiments under random data missing scenarios on datasets like MOSI, MOSEI, and SIMS demonstrate LNLN’s superior performance. The method ensures quality representations of the dominant modality and offers improved uniformity, transparency, and fairness in evaluations, consistently outperforming existing baselines across various challenging metrics.

**Strengths:**

Strength:
1. The paper addresses the issue of incomplete data in multimodal sentiment analysis, a prevalent and valuable research problem in real-world applications.

2. The manuscript is well-structured, with a clear and logical flow of ideas. The organization facilitates understanding and ensures that the arguments and methodologies are presented in a coherent manner.

**Weaknesses:**

Weakness：

1. The contributions summarized in the introduction are not clearly articulated. Additionally, using language as the primary information carrier to reconstruct incomplete data is a concept frequently employed in multimodal sentiment analysis (MSA). What sets your method apart from other similar approaches?

2. The so-called "Adaptive Hyper-modality Learning" essentially involves using transformers to facilitate interactions between different modalities. This interaction process appears to lack algorithmic innovation and seems to adopt a rather naive approach.

3. The primary structure of the Reconstructor is also based on transformers. Why did you choose transformers for reconstruction?

4. The entire algorithm's architecture heavily relies on transformers, but the transformers used in different modules have not been specifically tailored to the characteristics of this task.

5. In the experimental section, although LNLN demonstrates generally good performance, there is no in-depth comparative analysis with the most similar baseline, ALMT. Since LNLN is an improvement based on ALMT, such a detailed analysis is essential.

6. Ablation studies and other experiments were only conducted on the MOSI dataset. Why were these not performed on all datasets?

7. The experimental section lacks visualizations or case study to demonstrate the effectiveness of the algorithm, making it difficult to fully substantiate the algorithm's efficacy.

**Questions:**

Weakness：

1. The contributions summarized in the introduction are not clearly articulated. Additionally, using language as the primary information carrier to reconstruct incomplete data is a concept frequently employed in multimodal sentiment analysis (MSA). What sets your method apart from other similar approaches?

2. The so-called "Adaptive Hyper-modality Learning" essentially involves using transformers to facilitate interactions between different modalities. This interaction process appears to lack algorithmic innovation and seems to adopt a rather naive approach.

3. The primary structure of the Reconstructor is also based on transformers. Why did you choose transformers for reconstruction?

4. The entire algorithm's architecture heavily relies on transformers, but the transformers used in different modules have not been specifically tailored to the characteristics of this task.

5. In the experimental section, although LNLN demonstrates generally good performance, there is no in-depth comparative analysis with the most similar baseline, ALMT. Since LNLN is an improvement based on ALMT, such a detailed analysis is essential.

6. Ablation studies and other experiments were only conducted on the MOSI dataset. Why were these not performed on all datasets?

7. The experimental section lacks visualizations or case study to demonstrate the effectiveness of the algorithm, making it difficult to fully substantiate the algorithm's efficacy.

**Limitations:**

The algorithm's structure is overly reliant on transformers, and there is a lack of explanation as to why transformers were chosen for various operations, especially in relation to the specific characteristics of this task. Additionally, the paper introduces multiple loss functions; however, the effectiveness and necessity of each loss function are not thoroughly justified or demonstrated. This raises questions about whether each loss is indeed contributing as intended to the overall performance of the model.

---

> ### Author Rebuttal · Authors · 2024-08-05
>
> # Response to Reviewer xa9F
> ## **Response to Limitations**
> **Regarding the Use of Transformer:**
>
> **First,** our primary focus is on the innovation of the algorithm, not on the innovation of the Transformer. We have utilized Transformer layers in a similar manner to how CNNs and LSTMs are commonly employed in various models. It is only a tool to effectively model sequence data. Extensive MSA-related works [2, 4, 6, 7, 8, 10, 11] has demonstrated the efficacy of Transformers in processing sequential data. Thus, we chose to design the LNLN framework based on the Transformer structure. **Second**, to address concerns about the algorithm’s reliance on Transformers, we evaluated our model by replacing all Transformer layers with LSTM and MLP layers, which are widely used in earlier MSA works [1, 3, 5]. **The Table below** shows that while there is a slight performance degradation, the overall performance remains competitive and still achieves SOTA results across many metrics. It also demonstrates that **our algorithm's structure is not overly reliant on transformers.** In addition, the use of Transformer to implement the reconstructor is also seen in the previous methods [4, 8], which is normal practice in this field. Our focus is on validating the effectiveness of our proposed hypothesis through the LNLN framework.
>
> |MOSI|||||||
> |-|-|-|-|-|-|-|
> |Methods|Acc-7|Acc-5|Acc-2|F1|MAE|Corr|
> |LNLN|**34.26**|**38.27**|**72.55/70.94**|**72.73/71.25**|**1.046**|**0.527**|
> |LNLN(LSTM+MLP)|33.81|38.26|70.86/69.43|70.85/69.52|1.088|0.499|
> |**MOSEI**|||||||
> |Methods|Acc-7|Acc-5|Acc-2|F1|MAE|Corr|
> |LNLN|**45.42**|**46.17**|**76.30/78.19**|**77.77/79.95**|**0.692**|**0.530**|
> |LNLN(LSTM+MLP)|43.48|44.28|76.05/77.18|77.59/79.32|0.716|0.517|
> |**SIMS**|||||||
> |Methods|Acc-5|Acc-3|Acc-2|F1|MAE|Corr|
> |LNLN|**34.64**|**57.14**|**72.73**|79.43|**0.514**|**0.397**|
> |LNLN(LSTM+MLP)|27.87|55.50|70.23|**80.57**|0.560|0.257|
>
> **Regarding the effectiveness and necessity of each loss function:**
>
> **As shown in Tab. 4 of the paper**, we have discussed the effectiveness of each loss. We have additionally conducted our experiments on MOSI and SIMS datasets. In addition, **as shown in Tab. 1 of the uploaded PDF**, the results show that our losses are effectiveness and necessity.
>
> ## **Response to W1/Q1**
> **Regarding the contributions**: Please see the general Response.
>
> **Regarding the use of language as the primary information carrier:**
>
> While language plays a crucial role in MSA, previous works [1, 2, 3, 4, 5, 6, 8, 9, 10, 11] have not fully emphasized the importance of the language modality in method design. Our proposed LNLN addresses this gap by introducing the DMC module, specifically designed to improve the robustness of language features under varying noise scenarios.
>
> **Regarding the differentiation from existing methods:**
>
> 1. Unlike many existing MSA methods [1, 2, 3, 4, 5, 6, 8, 9, 10, 11] that treat all modalities equally, our method prioritizes the integrity of the dominant modality (language). We propose that by ensuring the completeness and quality of this modality, the overall robustness of the model can be significantly improved.
> 2. Our method uniquely incorporates a Completeness Check step within the DMC module, which evaluate and adjusts the representation of the dominant modality to maintain its quality. This method is distinct from other methods [1, 2, 3, 4, 5, 6, 7, 8, 9, 10, 11] used in MSA, which do not emphasize the completeness of the dominant modality to this extent.
> 3. Our model is specifically designed to tackle a wide range of noise levels, reflecting real-world scenarios where data quality can vary. The ability to maintain robust performance across these conditions is a key differentiator of our approach compared to other methods that may only perform well under limited or ideal scenarios.
>
> ## **Response to W2/Q2, W3/Q3 and W4/Q4**
> **As mentioned in the above**, our primary goal is not to introduce a novel Transformer architecture but to explore how to achieve robust MSA. The innovation of our LNLN lies in the overall framework, particularly in the design of the DMC module. The collaboration between AHL and DMC ensures that the model can effectively mitigate the impact of noise, thus achieving robust MSA. For more details, please refer to the Response to Limitations above.
>
> ## **Response to W5/Q5 and W7/Q7**
> **Tab. 8 and Tab. 9 of the paper** have detailed the performance of each method under the influence of different noise intensities, demonstrating the superiority of our method compared to ALMT. In addition, **as shown in Fig. 1 and Fig. 2 of the uploaded PDF**, we have compared the confusion matrices of ALMT and LNLN, and plotted their F1 variation curves under different noise intensities. It is clear that ALMT does not perform as well as LNLN in high-noise scenarios, especially when it also tends to favour almost all of its predictions to specific classes. As **shown** **in Fig. 3 of the uploaded PDF,** We conducted a Case Study through visualisation, and we can see that our LNLN is able to perceive the sentiment cues accurately for the hard samples and thus make accurate predictions. Due to word limitations of rebuttal, we will provide a more detailed analysis in the revised version.
>
> ## **Response to W6/Q6**
> We chose to perform our ablation studies on the MOSI dataset because it is widely used in previous works for ablation studies [1, 2, 3, 4, 5, 6, 7, 8, 9, 10, 11]. This allowed us to maintain consistency with existing works. Additionally, due to time limitation of rebuttal, we have tried our best to conduct the ablation studies on the SIMS dataset. The SIMS dataset is a Chinese dataset which differs significantly from MOSI. It allows us to demonstrate the generalizability and robustness of our approach across different languages and cultural contexts. **As shown in Tab. 4 of the uploaded PDF,** the result highlights the effectiveness of our method on this diverse dataset.

---

> > ### Comment · Reviewer_xa9F · 2024-08-09
> >
> > Thank you for your response and clarifications. I have adjusted my rating to 7.

---

> > > ### Author Response · Authors · 2024-08-09
> > >
> > > Thank you for adjusting your rating and for your support of our work. The feedback is important. We are grateful for the opportunity to improve our paper, and appreciate your time and effort.

---

### Official Review · Reviewer_ugFZ · 2024-07-04

**Soundness:** 2
**Presentation:** 3
**Contribution:** 3
**Rating:** 4
**Confidence:** 4

**Summary:**

This paper presents LNLN, which aims to address the challenge of data incompleteness in real-world scenarios caused by sensor failures or automatic speech recognition issues. The core idea is that even if other modalities are missing, the system can still work if the information from the dominant modality is complete. Therefore, LNLN aims to augment the dominant modality (text). By focusing on text, the method uses other modalities to reconstruct missing textual information, thereby enhancing robustness.

Core Contributions:

1. Evaluation of MSA methods with missing data.
2. The LNLN method.

**Strengths:**

The paper introduces LNLN to improve the robustness of multimodal sentiment analysis (MSA) by reinforcing the language modality. The Dominant Modality Correction (DMC) module and Dominant Modality-based Multimodal Learning (DMML) module enhance the model's robustness in various noisy scenarios by ensuring the quality of dominant modality representations.

Comprehensive experiments are conducted on multiple datasets (e.g., MOSI, MOSEI, and SIMS), simulating random data missing scenarios, and comparing with existing methods. The method provides additional consistency and achieves good performance.

**Weaknesses:**

The evaluation is indeed useful and addresses the gap in existing MSA methods with missing data. However, the innovation is limited and not sufficient to pave the way for future research, resulting in a modest contribution.

The performance improvement is not significant, and in some cases, it does not achieve state-of-the-art (SOTA) results.

Based on my understanding, Section 3.4 is essentially an application of GAN, which does not highlight significant innovation.

**Questions:**

Should the citation format be "cite" instead of "citet"?

In line 45, is your hypothesis supported by analysis, or is it first hypothesized and then experimentally validated?

How does the Completeness Check in line 168 work, and what role does the randomly initialized $H_{cc}$ play?

**Limitations:**

See weakness and question

---

> ### Author Rebuttal · Authors · 2024-08-05
>
> # Response to Reviewer ugFZ
>
> ## **Response to W1**
>
> We believe that our contributions are significant and address the gap in current MSA research. The robustness of MSA models in real-world, noisy environments is an important area of study. Most previous methods [1, 2, 3, 4, 5, 6, 7, 11] are evaluated on standard datasets, which may not accurately represent the challenges posed by real-world data. While some recent works [8, 9, 10] have attempted to explore robust MSA in noisy environments, these efforts often selectively report partial metrics, such as Acc-2 and F1, leading to the performance of various MSA methods in noisy scenarios under-explored. Our work aims to bridge this gap by providing a comprehensive evaluation of advanced MSA methods under noisy conditions, which we believe is a crucial step toward making these models more applicable to real-world scenarios. During this evaluation, we identified several overlooked issues. For example, our evaluation revealed a critical but neglected issue: existing models generally fail under high noise conditions. In such cases, most predictions tend to favor the more numerous categories, making the model appear to achieve SOTA results. However, this behavior is misleading and does not reflect the true performance of the model, leading to an unfair comparison with other approaches. **More details can be seen in Fig. 3, Fig. 4 and Fig. 5 in the appendix of the paper, and Fig. 1 of the uploaded PDF.**
>
> ## **Response to W2**
>
> We would like to clarify and emphasize the following points to address your concern:
>
> 1. **As shown in Tab. 1 and Tab. 2 of the paper**, we believe that our method outperforms existing methods in several key aspects. For example, on the MOSI dataset, our method achieves significant improvements in challenging metrics such as Acc-7 and Acc-5, with relative increases of 9.46% and 2.74%, respectively.
> 2. The reason for not achieving significant improvements in some cases is that our overall performance results are averaged across different noise levels. As mentioned in the **Response to W1** and in **Section 4.3 of the paper**, some metrics did not show significant improvement. This is because that the predictions of other models were heavily biased towards the more numerous categories under high noise scenarios, resulting those predictions meaningless. **For example**, in **Tab. 8 in the appendix of the paper**, when the data missing rate is 0.9, the Acc-7 of TFR-Net, which is specifically designed for data-missing scenarios, is 41.73%. It is slightly higher than that of LNLN at 40.10%. However, as shown in the confusion matrix in **Fig. 4 of the appendix**, TFR-Net predicted almost all test samples to be in Category 3 (i.e., Neutral). This bias significantly inflated the overall performance of TFR-Net in **Tab. 1 of the paper**, but these predictions are not meaningful in a high-noise scenario. This issue is one reason why TFR-Net's performance appears higher, but it does not reflect genuine robustness.
> 3. As mentioned in **Response to W1,** we aimed to provide a fair and comprehensive evaluation, reporting as many relevant metrics as possible rather than selectively reporting only favorable results. It is unrealistic to expect a model to show significant improvements across all fine-grained metrics simultaneously. However, our method consistently maintains a competitive performance level across all reported metrics, highlighting its robustness.
>
> ## **Response to W3**
>
> We would like to clarify that while adversarial learning is employed as a tool within our model, the innovation lies in how we utilize it to address specific challenges in MSA. Similar to how other works employ CNNs and LSTMs as basic layers, we have chosen this pipeline to achieve our specific objectives. The innovation is embodied in the design of the DMC module. This pipeline is carefully designed based on our hypothesis and offers a novel approach to improving the robustness in MSA. Unlike previous approaches [8, 9] that typically treat each modality equally and reconstruct whichever information is missing, our method prioritizes the dominant modality (language) and focuses on improving its completeness in noisy scenarios. Overall, our innovative application of adversarial learning is not just in the use of GANs, but in how it is integrated into our approach to address the specific challenges of robust MSA.
>
> ## **Response to Q1**
>
> As to the format of the references themselves, any style is acceptable as long as it is used consistently. You can refer to the PDF of "Formatting Instructions For NeurIPS 2024".
>
> ## **Response to Q2**
>
> The hypothesis is first proposed and then experimentally validated.
>
> ## **Response to Q3**
>
> The Completeness Check mainly used to evaluate the c*ompleteness* of the language features and determine to what extent these features are affected by noise or missing data.
>
> 1. $H_{cc}$ is a randomly initialized token used as an initial embedding that is concatenated with language features $H_l^1$. Through learning, it is used to predict the completeness of the $H_l^1$. Random initialization ensures that the feature is free from any a priori bias.
> 2. First, the language representation $H^1_l$ and $H_{cc}$ are fed into the encoder. The encoder processes the input and capture the completeness of the $H^1_l$. Then, the encoder outputs a completeness weight ($w$) that indicates the degree of completeness. This weight is used to balance the $H^1_l$ and the proxy features $H^1_p$ generated by the adversarial learning. Finally, the $w$ is used to combine the $H^1_l$ with the generated proxy features. This ensures that the final dominant modality representation maintains high quality and integrity in the presence of noise.
> 3. By evaluating and adjusting the dominant modality representation through this process, the Completeness Check helps maintain the robustness and reliability of the model.

---

### Author Rebuttal · Authors · 2024-08-05

# General Response

Dear Reviewers, ACs and SACs,

We would like to express our sincere gratitude for your thoughtful questions and valuable feedback. We greatly appreciate the time and effort you have invested in reviewing our paper. **We are eager to engage in further discussions with you to address your concerns and enhance the quality of our work.**

**First, we would like to restate our contributions:**

1. We provide an extensive evaluation of existing methods, offering a deeper understanding of the performance of current advanced methods in noise scenarios.
2. Our LNLN is designed to improve the overall robustness of MSA by ensuring the quality of the dominant modality (language) across various levels of noise. This is achieved through the novel Dominant Modality Correction (DMC) pipeline, which offers new insights into addressing noise in MSA.
3. Extensive experiments and ablation studies demonstrate the effectiveness of LNLN, showing that it achieves state-of-the-art performance across most metrics, particularly under high levels of noise.

**To better respond to your questions, we have conducted several additional experiments:**

1. We conducted ablation studies on the SIMS dataset, including the model’s dependence on the Transformer structure (**see the Table in Response to Limitations of Reviewer xa9F**), effectiveness and necessity of each loss (**see Tab. 1 of the uploaded PDF**), effects of different regularization on the SIMS dataset (**see Tab. 2 of the uploaded PDF**), and the effect of different component on the SIMS dataset (**see Tab. 2 of the uploaded PDF**).
2. We included more comparisons with previous methods, incorporating visualization (**see Fig. 1 and Fig. 2 of the uploaded PDF**) and a case study (**see Fig. 3 of the uploaded PDF**), with a particular focus on the suboptimal model ALMT.

Specifically, **for the Table in Response to Limitations of Reviewer xa9F**, we replaced the Transformer with LSTM and MLP to build the LNLN model. Although there was a slight decrease in accuracy, the performance still surpassed other advanced methods. This demonstrates that our proposed algorithm does not heavily rely on the Transformer architecture. **For Tab. 1 of the uploaded PDF,** we provided a detailed analysis of the effectiveness of each loss component on the MOSI and SIMS datasets. The results confirm that each part of the loss function is beneficial and contributes to the overall performance of our algorithm. **For Tab. 2 of the uploaded PDF,** we presented ablation results of LNLN on the SIMS dataset. The SIMS dataset, as a Chinese dataset, is quite different from MOSI. The results demonstrate that LNLN has strong generalization capabilities, further demonstrating the effectiveness of our method. **For Fig. 1 of the uploaded PDF,** we compared the confusion matrices of LNLN and ALMT across the three datasets. Consistent with the results shown in **Fig. 3, Fig. 4, and Fig. 5 in the Appendix of the paper**, ALMT suffers from severe invalid predictions in high-noise scenarios, where most samples are predicted to fall into a particular category. Although our proposed LNLN also exhibits this phenomenon to some extent, it is not as severe as with ALMT. This further demonstrates the advantage of our method in processing noise data. **For Fig. 2 of the uploaded PDF,** we compared the F1 accuracy curves of several advanced methods across different random missing rates. The curve shows that LNLN maintains SOTA performance across various random missing rates, especially in high-noise scenarios, demonstrating the robustness of our method. **For Fig. 3 of the uploaded PDF,** we show case studies. In scenarios where key information is masked, such as in the text of Figure (a), our LNLN model can still predict correctly. In Figure (b), both ALMT and LNLN predict incorrectly, highlighting the challenges the model faces in extremely noisy conditions.

Thank you for your patience and support. **We look forward to continuing the discussion and refining our work with your valuable** **insights****.**

Sincerely,

The Authors

## **Reference Used throughout the Rebuttal**

[1] Zadeh, A., Chen, M., Poria, S., Cambria, E., Morency, L., 2017. Tensor fusion network for multimodal sentiment analysis, in EMNLP.

[2] Tsai, Y.H., Bai, S., Liang, P.P., Kolter, J.Z., Morency, L., Salakhutdinov, R., 2019. Multimodal transformer for unaligned multimodal language sequences, in ACL.

[3] Hazarika, D., Zimmermann, R., Poria, S., 2020. MISA: modality-invariant and-specific representations for multimodal sentiment analysis, in ACM MM.

[4] Liang, J., Li, R., Jin, Q., 2020. Semi-supervised multi-modal emotion recognition with cross-modal distribution matching, in ACM MM.

[5] Yu, W., Xu, H., Yuan, Z., Wu, J., 2021. Learning modality-specific representations with self supervised multi-task learning for multimodal sentiment analysis, in AAAI.

[6] Yang, D., Huang, S., Kuang, H., Du, Y., Zhang, L., 2022. Disentangled representation learning for multimodal emotion recognition, in ACM MM.

[7] Zhang, H., Wang, Y., Yin, G., Liu, K., Liu, Y., Yu, T., 2023. Learning language-guided adaptive hyper-modality representation for multimodal sentiment analysis, in EMNLP.

[8] Yuan, Z., Li, W., Xu, H., Yu, W., 2021. Transformer-based feature reconstruction network for robust multimodal sentiment analysis, in ACM MM.

[9] Yuan, Z., Liu, Y., Xu, H., Gao, K., 2024. Noise imitation based adversarial training for robust multimodal sentiment analysis. IEEE TMM.

[10] Li, M., Yang, D., Lei, Y., Wang, S., Wang, S., Su, L., Yang, K., Wang, Y., Sun, M., Zhang, L., 2024. Aunified self-distillation framework for multimodal sentiment analysis with uncertain missing modalities, in AAAI.

[11] Lv, F., Chen, X., Huang, Y., Duan, L., Lin, G., 2021. Progressive modality reinforcement for human multimodal emotion recognition from unaligned multimodal sequences, in CVPR.

## **Rebuttal PDF**

---

### Author Response · Authors · 2024-08-14

Dear All,

We would like to express our sincere thanks for the time and effort you have invested in reviewing our paper. We have responded to all of the comments and questions raised during the review process and hope that our responses address your concerns adequately.

Thank you for your valuable feedback.

Best regards,

The Authors

---

### Decision · Program_Chairs · 2024-09-25

**Decision:**

Accept (poster)

**Comment:**

The paper introduces the Language-dominated Noise-resistant Learning Network (LNLN) for robust Multimodal Sentiment Analysis, designed to address the challenge of incomplete data in real-world applications. LNLN leverages a Dominant Modality Correction module and a Dominant Modality-based Multimodal Learning module to enhance the robustness of sentiment analysis by focusing on the integrity of the language modality, which typically contains the most dense sentiment information. The model is tested across various datasets with random missing data scenarios, consistently outperforming existing methods.

The paper is well-written and easy to follow. The reviewers had some questions and concerns about the evaluation, datasets used in the experiments, technical details, and the motivation behind the paper. During the rebuttal, the authors addressed these issues, and the reviewers were satisfied with the responses and additional experiments. Therefore, I recommend accepting the paper.